# Weaker MVI Condition: Extragradient Methods with Multi-Step Exploration

**Yifeng Fan**    **Yongqiang Li**    **Bo Chen**
Zhejiang University of Technology
`{yifengfan,yqli,bchen}@zjut.edu.cn`

## Abstract

This paper proposes a new framework of algorithms that is extended from the celebrated extragradient algorithm. The min-max problem has attracted increasing attention because of its applications in machine learning tasks such as generative adversarial networks (GANs) training. While there has been exhaustive research on convex-concave setting, problem of nonconvex-nonconcave setting faces many challenges, such as convergence to limit cycles. Given that general min-max optimization has been found to be intractable, recent research efforts have shifted towards tackling structured problems. One of these follows the *weak Minty variational inequality* (weak MVI), which is motivated by relaxing *Minty variational inequality* (MVI) without compromising convergence guarantee of extragradient algorithm. Existing extragradient-type algorithms involve one exploration step and one update step per iteration. We analyze the algorithms with multiple exploration steps and show that current assumption can be further relaxed when more exploration is introduced. Furthermore, we design an adaptive algorithm that explores until the optimal improvement is achieved. This process exploits information from the whole trajectory and effectively tackles cyclic behaviors.

## 1 Introduction

Min-max optimization has aroused recent interest due to its significant applications in machine learning tasks, such as generative adversarial networks, adversarial training, robust learning and sharpness-aware minimization (Goodfellow et al., 2014; Madry et al., 2018; Levy et al., 2020; Foret et al., 2020).

Min-max optimization aims to find saddle point of a objective function. Theories and methods have been extensively studied for decades, usually through the prospective of variational inequalities. There has been a plethora of literature focusing on convex-concave (Tseng, 2008; Nesterov, 2007; Nemirovski, 2004) or nonconvex-concave (Xu et al., 2023; Boţ & Böhm, 2023; Lin et al., 2020) objectives. However, problem with nonconvex-nonconcave objective remains relatively obscure. Finding local solution in general nonconvex-nonconcave problems has been proved intractable (Daskalakis et al., 2021).

One powerful tool that has been revisited is the celebrated extragradient (Korpelevich, 1976), which has been found effective in solving a class of nonconvex-nonconcave problems (Daskalakis et al., 2020). Diakonikolas et al. (2021) first introduce this new structure as weak Minty variational inequality, featuring a weaker assumption than monotonicity. They make a slight modification on extragradient (named EG+) and establish convergence guarantee on this structure. Pethick et al. (2022) elucidate the mechanism of EG+ and prove a tight parameter range of $\rho > -1/2L$, where $\rho$ represents the weak Minty parameter that describes the degree of non-monotonicity and $L$ is the Lipschitz constant, which is the known best range to have convergence guarantee for this class of problems.

**Our contributions**    Building on the works of Pethick et al. (2022) and Diakonikolas et al. (2021), we generalize the extragradient algorithm to multi-step cases which adapt to a larger range of problems. Furthermore, we propose a new algorithm that exploits more than local information by

introducing adaptive exploration.

1. We analyze the algorithms with two and more exploration steps. We discover that the range of $\rho$ in weak MVI assumption can be relaxed when more exploration steps are introduced. Especially, when the sum of exploration stepsizes in one iteration is still bounded by $1/L$, a similar type of convergence guarantee can be established. Under this restriction, the assumption parameter can be relaxed to $\rho > -(1-1/e)/L$. Additionally, for $n = 2$, we establish the range of $\rho$ where the stepsizes are unrestricted by these constraints.

2. Building on the established convergence results, we introduce a novel algorithm that adaptively increases the number of exploration steps. This is inspired by the idea of interpreting each update as a projection onto a certain hyperplane, defined by the weak Minty inequality. Every exploration point provides information and helps narrow down the target range. This algorithm effectively tackles cyclic behaviors.

**Related work**   Recent application in GAN training has motivated research on min-max optimization and revisit to classic algorithms. A line of pioneering works introduces the traditional perspective of variational inequality and resorts to optimistic methods (Daskalakis et al., 2018; Gidel et al., 2018; Mertikopoulos et al., 2018). Simple algorithm such as gradient descent ascent is also examined under the Polyak-Łojasiewicz condition (Yang et al., 2020). Hsieh et al. (2021) revisit a list of algorithms including extragradient and demonstrate that current methods still have trouble avoiding limit cycles.

Weak Minty variational inequality, also known as star-negative comonotonicity, has received recent attention from community and has become a common structure setting. Diakonikolas et al. (2021) first introduced weak MVI as an important assumption, under which their EG+ algorithm enjoys $\mathcal{O}(1/\sqrt{k})$ convergence rate. Pethick et al. (2022) provide a full picture of the algorithm with an adaptive stepsize selection technique. They also propose an algorithm with backtracking line search incorporated, which effectively escape limit cycles and outperforms existing methods. Böhm (2022) propose another variant of EG+ with adaptive stepsize, not requiring knowledge of problem parameter $\rho$ and $L$. While most literature provide best-iterate convergence results, last-iterate convergence of EG+ under weak MVI has also been established (Gorbunov et al., 2023; Tran-Dinh, 2023), but under more restrictive parameter $\rho > -1/8L$.

Another line of work resorts to anchoring techniques. Lee & Kim (2021) combine anchored extragradient with separate stepsizes and obtain a fast $\mathcal{O}(1/k)$ rate on gradient norm. Alcala et al. (2023) introduce new moving anchor technique generalizing current algorithms and attain optimal convergence rate. These works are under a more restrictive (unstarred) negative comonotonicity assumption.

When it comes to stochastic methods, similar thought to EG+ can be found in Hsieh et al. (2020), where they show that doubled exploration stepsize in stochastic extragradient is effective on avoiding cycles. Diakonikolas et al. (2021) extend their deterministic result to stochastic setting with unbiased oracle of gradient and bounded variance by increasing oracle queries. Pethick et al. (2023) establish almost sure convergence with single query per iteration, involving one fixed and one diminishing stepsize. Their method works under the same assumption of $\rho > -1/2L$. Aside from separate stepsizes, other variants such as stochastic past extragradient (SPEG) have been shown effective under weak MVI assumption, along with a new *expected residual condition* (Choudhury et al., 2023).

## 2   PRELIMINARIES

We consider two real vectors, $\boldsymbol{x} \in \mathbb{R}^{d_x}, \boldsymbol{y} \in \mathbb{R}^{d_y}$ and minimax problems of the form:

$$\min_{\boldsymbol{x} \in \mathbb{R}^{d_x}} \max_{\boldsymbol{y} \in \mathbb{R}^{d_y}} f(\boldsymbol{x}, \boldsymbol{y}) \tag{2.1}$$

where $f : \mathbb{R}^{d_x} \times \mathbb{R}^{d_y} \to \mathbb{R}$ is a smooth (possibly nonconvex-nonconcave) function and $d_x + d_y = d$.

To study the stationary point, we consider the saddle gradient operator $F : \mathbb{R}^d \to \mathbb{R}^d$ defined via $F\boldsymbol{z} = \begin{bmatrix} \nabla_{\boldsymbol{x}} f(\boldsymbol{x}, \boldsymbol{y}) \\ -\nabla_{\boldsymbol{y}} f(\boldsymbol{x}, \boldsymbol{y}) \end{bmatrix}$, where $\boldsymbol{z} = \begin{bmatrix} \boldsymbol{x} \\ \boldsymbol{y} \end{bmatrix}$.

We want to find zeros of this operator, the set of which denoted by $\mathbf{zer}F := \{z \in \mathbb{R}^d \mid Fz = 0\}$. This is a first-order necessary condition of $z$ being the saddle point satisfying (2.1).

In this paper, we study problems in which operator $F$ satisfies the following assumptions.

**Assumption 1.** (*L-Lipschitz continuity*). *Operator F is L-Lipschitz continuous. For any $u, v \in \mathbb{R}^d$,*

$$\|Fu - Fv\| \le L\|u - v\| \tag{2.2}$$

**Assumption 2.** (Weak MVI). *There exists $z^* \in \mathbf{zer}F$ such that for any $z \in \mathbb{R}^d$,*

$$\langle Fz, z - z^* \rangle \ge \rho\|Fz\|^2 \tag{2.3}$$

*for some $\rho \in (\rho_0, \infty)$.*

In this paper, the proposed methods provide convergence guarantee for problems where $\rho \in (-^{(1-1/e)}/_L, \infty)$, where $e$ is Euler's number.

### 2.1 PRELIMINARY ALGORITHMS

**EG** (Korpelevich, 1976):

$$\begin{aligned}
\bar{z}^k &= z^k - \alpha_k Fz^k \\
z^{k+1} &= z^k - \alpha_k F\bar{z}^k
\end{aligned} \tag{EG}$$

where $\gamma_k$ is the stepsize.

Extragradient is a classical algorithm for saddle point problems. Often regarded as an explicit approximation of the proximal point method, EG employs the same stepsize for both extrapolation and update phases.

**EG+** (Diakonikolas et al., 2021):

$$\begin{aligned}
\bar{z}^k &= z^k - \gamma_k Fz^k \\
z^{k+1} &= z^k - \alpha_k F\bar{z}^k
\end{aligned} \tag{EG+}$$

where $\gamma_k = {}^1/_L$ and $\alpha_k = {}^1/_{2L}$ are the stepsizes.

EG+ is a generalization of EG, allowing an aggressive extrapolation step. This slight modification makes it effective for weak MVI problems under $\rho > -^1/_{8L}$.

**AdaptiveEG+** (Pethick et al., 2022):

$$\begin{aligned}
\bar{z}^k &= z^k - \gamma_k Fz^k \\
\alpha_k &= \sigma_k - \frac{\langle F\bar{z}^k, \bar{z}^k - z^k \rangle}{\|F\bar{z}^k\|^2} \\
z^{k+1} &= z^k - \lambda_k \alpha_k F\bar{z}^k, \lambda_k \in (0, 2)
\end{aligned} \tag{AdaptiveEG+}$$

where $\gamma_k \in (\lfloor -2\rho \rfloor_+, {}^1/_L]$, $\alpha_k$ are the stepsizes and $\sigma_k \in (-\gamma_k/2, \rho]$, $\lambda_k \in (0, 2)$ are relaxation parameters.

This algorithm provides a tight range of $\alpha_k$ for convergence of (EG+). Diving into the conception of projection, (AdaptiveEG+) broadens the problem range to $\rho > -^1/_{2L}$, which has been the best known result for weak MVI problems.

### 2.2 PRELIMINARY DEFINITIONS

**Definition 2.1.** (*Weak MVI halfspace*). *Given $u \in \mathbb{R}^d$. Define weak MVI halfspace at $u$ as the set restricted by (2.3) at $u$,*

$$\mathcal{D}(u) := \{w \in \mathbb{R}^d \mid \langle Fu, u - w \rangle \ge \rho\|Fu\|^2\} \tag{2.4}$$

*The boundary of $\mathcal{D}(u)$ is a hyperplane with $Fu$ as normal vector,*

$$\partial\mathcal{D}(u) := \{w \in \mathbb{R}^d \mid \langle Fu, u - w \rangle = \rho\|Fu\|^2\} \tag{2.5}$$

Under Assumption 2, we have $\forall \boldsymbol{u} \in \mathbb{R}^d, \boldsymbol{z}^* \in \mathcal{D}(\boldsymbol{u})$. As we access gradient at any point, we get the information that zero $\boldsymbol{z}^*$ must be inside the corresponding halfspace.

**Definition 2.2.** (Signed distance). *Given $\boldsymbol{u} \in \mathbb{R}^d$ and convex set $\mathcal{S} \subset \mathbb{R}^d$. The signed distance from $\boldsymbol{u}$ to $\mathcal{S}$ is defined by*

$$d(\boldsymbol{u}, \mathcal{S}) := \begin{cases} d(\boldsymbol{u}, \partial \mathcal{S}), & \text{if } \boldsymbol{u} \in \mathcal{S}^c \\ -d(\boldsymbol{u}, \partial \mathcal{S}), & \text{if } \boldsymbol{u} \in \mathcal{S} \end{cases} \tag{2.6}$$

*where $\partial \mathcal{S}$ denotes the boundary of $\mathcal{S}$ and $\mathcal{S}^c$ denotes the complement of $\mathcal{S}$. $d(\boldsymbol{u}, \partial \mathcal{S})$ is defined by $d(\boldsymbol{u}, \partial \mathcal{S}) := \inf_{\boldsymbol{v} \in \partial \mathcal{S}} d(\boldsymbol{u}, \boldsymbol{v})$.*

The signed distance $d(\boldsymbol{u}, \mathcal{S})$ is positive when $\boldsymbol{u}$ is out of $\mathcal{S}$ and negative when $\boldsymbol{u}$ is inside $\mathcal{S}$.

In this paper, we focus on the signed distance between a point and a weak MVI halfspace, which is linear and therefore convex. More specifically, the iteration point $\boldsymbol{z}^k$ and weak MVI halfspace at exploration point $\boldsymbol{z}_i^k$.

**Lemma 2.3.** *Given $\boldsymbol{u}, \boldsymbol{v} \in \mathbb{R}^d$, the signed distance from $\boldsymbol{v}$ to $\mathcal{D}(\boldsymbol{u})$ is*

$$d(\boldsymbol{v}, \mathcal{D}(\boldsymbol{u})) = \frac{\rho \|F\boldsymbol{u}\|^2 - \langle F\boldsymbol{u}, \boldsymbol{u} - \boldsymbol{v} \rangle}{\|F\boldsymbol{u}\|} \tag{2.7}$$

## 3    GENERALIZED FRAMEWORK OF PROJECTION ALGORITHMS

Projection-type algorithms have been extensively utilized in variational inequality problems (Solodov & Tseng, 1996; Solodov & Svaiter, 1999). Recent work of Pethick et al. (2022) employs projection technique in generalizing extragradient algorithm.

Notice that in (AdaptiveEG+), the update stepsize $\alpha_k = \sigma_k - \frac{\langle F\bar{\boldsymbol{z}}^k, \bar{\boldsymbol{z}}^k - \boldsymbol{z}^k \rangle}{\|F\bar{\boldsymbol{z}}^k\|^2} \leq \frac{d(\boldsymbol{z}^k, \mathcal{D}(\bar{\boldsymbol{z}}^k))}{\|F\bar{\boldsymbol{z}}^k\|}$.

The main idea lies in perceiving each update as a projection onto the hyperplane $\partial \mathcal{D}(\bar{\boldsymbol{z}}^k)$. Due to Assumption 2, $\boldsymbol{z}^* \in \mathcal{D}(\bar{\boldsymbol{z}}^k)$, while $\boldsymbol{z}^k \notin \mathcal{D}(\bar{\boldsymbol{z}}^k)$ since the algorithm is designed to make $\alpha_k > 0$. As a result, the hyperplane naturally separates $\boldsymbol{z}^k$ and $\boldsymbol{z}^*$. The parameters of $\sigma_k \leq \rho$ and $\lambda \in (0, 2)$ bear no effect on convergence, given the fact that scaling the projection distance no more than twice still takes you closer to the hyperplane.

It is interesting to notice that in this process, there is no restriction on the selection of $\bar{\boldsymbol{z}}^k$. Literally any $\bar{\boldsymbol{z}}^k$ generates a halfspace that $\boldsymbol{z}^* \in \mathcal{D}(\bar{\boldsymbol{z}}^k)$. Therefore the same mechanism still applies as long as $\bar{\boldsymbol{z}}^k$ is chosen such that $\boldsymbol{z}^k \notin \mathcal{D}(\bar{\boldsymbol{z}}^k)$, or equivalently $d(\boldsymbol{z}^k, \mathcal{D}(\bar{\boldsymbol{z}}^k)) > 0$. In other words, as long as at every iteration point, we can find another point such that the iteration point is out of its weak MVI halfspace. The border hyperplane consequently separates the iteration point $\boldsymbol{z}^k$ and the desired zero $\boldsymbol{z}^*$. Therefore intuitively, a projection onto the hyperplane get closer to $\boldsymbol{z}^*$. This point does not necessarily have to be attained by a single forward operator evaluation $\bar{\boldsymbol{z}}^k = \boldsymbol{z}^k - \gamma_k F\boldsymbol{z}^k$.

Given the foregoing discussion, it is straightforward to propose the following framework for solving the weak MVI type saddle point problems.

$$\begin{aligned} \bar{\boldsymbol{z}}^k &= G_k \boldsymbol{z}^k \\ \alpha_k &= \sigma_k - \frac{\langle F\bar{\boldsymbol{z}}^k, \bar{\boldsymbol{z}}^k - \boldsymbol{z}^k \rangle}{\|F\bar{\boldsymbol{z}}^k\|^2} > 0, \sigma_k \leq \rho \\ \boldsymbol{z}^{k+1} &= \bar{\boldsymbol{z}}^k - \lambda_k \alpha_k F\bar{\boldsymbol{z}}^k, \lambda_k \in (0, 2) \end{aligned} \tag{3.1}$$

where for all $k$, $G_k : \mathbb{R}^d \to \mathbb{R}^d$ is a map such that $\forall \boldsymbol{z} \in \mathbb{R}^d, d(\boldsymbol{z}, \mathcal{D}(G_k \boldsymbol{z})) > 0$ and $\sigma_k \leq \rho$ is selected such that $\alpha_k > 0$.

In other words, if for every $\boldsymbol{z}^k$ we are able to find $\bar{\boldsymbol{z}}_k$ so that $\boldsymbol{z}^k \notin \mathcal{D}(\bar{\boldsymbol{z}}_k)$, then the gradient $F\bar{\boldsymbol{z}}_k$ can be used for extragradient update. Our objective is to find such maps that guarantee this property. One example is $G_k = id - \gamma_k F$ with $\lfloor -\frac{\rho}{1+\rho L} \rfloor_+ < \gamma_k \leq 1/L$ used in EG+, where $id$ denotes identity operator and $\lfloor x \rfloor_+ := \max\{0, x\}$.

Such algorithms enjoy similar convergence guarantee to (AdaptiveEG+)((Pethick et al., 2022), Thm. 3.1), with an $\mathcal{O}(1/\sqrt{k})$ best-iterate convergence rate.

**Theorem 3.1.** *Let F be L-Lipschitz and satisfy weak Minty condition with $\rho$. Let $\lambda_k \in (0, 2)$, $\sigma_k \leq \rho$. Assume that $\liminf_{k \to \infty} \lambda_k(2 - \lambda_k) > 0$ and $\liminf_{k \to \infty} \alpha_k > 0$. Assume that for all $k$, $d(\boldsymbol{z}_k, \mathcal{D}(\bar{\boldsymbol{z}}^k)) > 0$. Assume that $\sigma_k$ is selected such that $\alpha_k > 0$. Consider the sequences $(\boldsymbol{z}^k)_{k \in \mathbb{N}}$ and $(\bar{\boldsymbol{z}}^k)_{k \in \mathbb{N}}$ generated by (3.1). Then,*

$$\min_{k=0,1,\dots,m} \|F\bar{\boldsymbol{z}}^k\|^2 \leq \frac{1}{\kappa(m+1)} \|\boldsymbol{z}^0 - \boldsymbol{z}^*\|^2 \tag{3.2}$$

*where $\kappa = \liminf_{k \to \infty} \lambda_k(2 - \lambda_k)\alpha_k^2$. Moreover, $(\bar{\boldsymbol{z}}^k)_{k \in \mathbb{N}}$ converges to $\boldsymbol{z}^*$.*

Note that no limitation on $\rho$ is mentioned in this framework. Actually, the range of manageable $\rho$ depends on the selection of $\sigma_k$ and the map $G_k$, which will be covered in the next section. More specifically, $\rho > \sup_{\bar{\boldsymbol{z}}^k} \frac{\langle F\bar{\boldsymbol{z}}^k, \bar{\boldsymbol{z}}^k - \boldsymbol{z}^k \rangle}{\|F\bar{\boldsymbol{z}}^k\|^2}$, where the right-hand side is related to the settings of $G_k$.

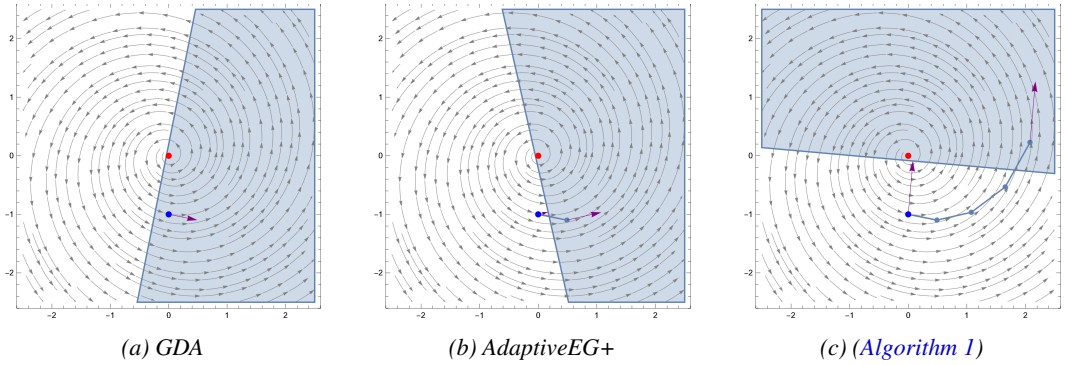

*(a) GDA*          *(b) AdaptiveEG+*          *(c) (Algorithm 1)*

*Figure 1: Intuition of projection algorithms*

From another perspective, every operator evaluation $F\boldsymbol{z}$ provides new information about $\boldsymbol{z}^*$. $\boldsymbol{z}^* \in \mathcal{D}(\boldsymbol{z})$ rules out the possibility that $\boldsymbol{z}^*$ is in another halfspace. This explains why increasing exploration plays a crucial role. Gradient descent ascent fails in $\rho < 0$ weak MVI problems since both $\boldsymbol{z}^k$ and $\boldsymbol{z}^*$ is in $\mathcal{D}(\boldsymbol{z}^k)$ and the hyperplane cannot separate them. (AdaptiveEG+) manages to find suitable $\bar{\boldsymbol{z}}^k$ with a larger extrapolation stepsize to separate them and update (project) with a smaller stepsize.

It is then natural to increase exploration by taking further steps in the subroutine $G_k$, pursuing a larger projection distance. See Fig. 1 for intuition. This naturally leads to a question: what kind of convergence guarantee can be provided for aforementioned multi-step algorithms? We address this problem in Section 4 and introduce the "max distance" algorithm in Section 5.

## 4 MULTI-STEP EXTRAGRADIENT

We start from the (AdaptiveEG+) algorithm and generalize it to multi-step cases.

Pethick et al. (2022) obtained a range of $\rho \in (-\frac{1}{2L}, \infty)$ for the algorithm to converge and demonstrated its tightness. Moreover, they restated the condition as $\rho > -\gamma_k/2$, where the stepsize satisfies $\gamma_k \leq 1/L$. We reexamine the claim and articulate our understanding.

The key is to assure a positive projection distance. Taking $\sigma_k = \rho$,

$$\alpha_k = \rho - \frac{\langle F\bar{\boldsymbol{z}}^k, \bar{\boldsymbol{z}}^k - \boldsymbol{z}^k \rangle}{\|F\bar{\boldsymbol{z}}^k\|^2} = \rho + \frac{\gamma_k \langle F\bar{\boldsymbol{z}}^k, F\boldsymbol{z}^k \rangle}{\|F\bar{\boldsymbol{z}}^k\|^2} \geq \rho + \frac{\gamma_k}{1 + \gamma_k L} \tag{4.1}$$

The inequality follows from rearranging $\|F\bar{\boldsymbol{z}}^k - F\boldsymbol{z}^k\| \leq \gamma_k L \|F\boldsymbol{z}^k\|$ into $\|F\boldsymbol{z}^k - \frac{1}{1 - \gamma_k^2 L^2} F\bar{\boldsymbol{z}}^k\| \leq \frac{\gamma_k L}{1 - \gamma_k^2 L^2} \|F\bar{\boldsymbol{z}}^k\|$ and applying Cauchy-Schwarz inequality on it.

Rather than the stated $\rho > -\gamma_k/2$ which only applies when $\gamma_k \leq 1/L$, a more precise condition should be $\rho > -\frac{\gamma_k}{1 + \gamma_k L}$. Increasing extrapolation stepsize $\gamma_k$ does not extend the lower bound of $\rho$ infinitely,

but push it closer to $-1/L$. Actually, objective functions with $\rho \leq -1/L$ are knotty for first order methods. For example, both $Fz = Lz$ and $Fz = -Lz$ falls into the structure. Particularly, when $\beta \leq -1$, $\beta$-cohypomonotone operator $F$ may fail to have an at most single-valued resolvent $J_F$, making it problematic in finding fixed points of $J_F$ (Table 1, (Bauschke et al., 2021)).

### 4.1 2-STEP EXTRAGRADIENT

Consider the $n$-step extragradient.

$$\forall i \in [n], \boldsymbol{z}_i^k = \boldsymbol{z}_{i-1}^k - \gamma_{k,i} F \boldsymbol{z}_{i-1}^k$$

$$\bar{\boldsymbol{z}}^k = \boldsymbol{z}_n^k$$

$$\alpha_k = \sigma_k - \frac{\langle F\bar{\boldsymbol{z}}^k, \bar{\boldsymbol{z}}^k - \boldsymbol{z}^k \rangle}{\|F\bar{\boldsymbol{z}}^k\|^2} > 0, \sigma_k \leq \rho \qquad \text{($n$-step EG)}$$

$$\boldsymbol{z}^{k+1} = \bar{\boldsymbol{z}}^k - \lambda_k \alpha_k F\bar{\boldsymbol{z}}^k, \lambda_k \in (0,2)$$

where $\boldsymbol{z}_0^k = \boldsymbol{z}^k$ is the current point, $\boldsymbol{z}_i^k, i \in [n]$ are intermediate steps, and $\boldsymbol{z}^{k+1}$ is the adopted update. The extrapolation point $\bar{\boldsymbol{z}}^k$ is attained by $n$ steps of gradient descent.

In this subsection we focus on the case where $n = 2$. Recall that selection of stepsizes and $\sigma_k$ plays a crucial rule in the algorithm and determines the range of problem parameter $\rho$ that the algorithm can address. We elaborate in the subsequent theorem how to choose parameters that ensures convergence.

**Theorem 4.1.** *Let F be L-Lipschitz and satisfy weak Minty condition with $\rho$. Let $n = 2$. Assume that for all $k$, $\gamma_{k,1} = \delta_1/L$, $\gamma_{k,2} = \delta_2/L$, and $\delta_1, \delta_2 \in (0,1)$. Assume that $\liminf_{k\to\infty} \lambda_k(2 - \lambda_k) > 0$ and $\liminf_{k\to\infty} \alpha_k > 0$. If for all $k$, $\sigma_k \leq \rho$ and*

$$\sigma_k > \begin{cases} -\frac{1}{L}\left[1 - \frac{1}{(1+\delta_1)(1+\delta_2)}\right] & \text{if } \delta_1 + \delta_2 \leq 1 \\ -\frac{1}{L}\left[\frac{\delta_1(1-\delta_1^2-\delta_2^2)}{2(1-\delta_1^2)(1-\delta_2^2)} + \frac{\delta_2}{1+\delta_2}\right] & \text{if } \delta_1 + \delta_2 > 1 \end{cases} \qquad (4.2)$$

*Then the sequence $(\bar{\boldsymbol{z}}^k)_{k\in\mathbb{N}}$ generated by ($n$-step EG) satisfies $\min_{k=0,1,\ldots,m}\|F\bar{\boldsymbol{z}}^k\|^2 \leq \frac{1}{\kappa(m+1)}\|\boldsymbol{z}^0 - \boldsymbol{z}^*\|^2$, where $\kappa = \liminf_{k\to\infty} \lambda_k(2 - \lambda_k)\alpha_k^2$.*

Selecting the parameters according to (4.2) guarantees best-iterate convergence. In the following theorem, we present the specific parameter selection that maximize the range of $\rho$ in both cases.

**Theorem 4.2.** *Let F be L-Lipschitz and satisfy weak Minty condition with $\rho$. Let $n = 2$. Assume that for all $k$, $\gamma_{k,1} = \delta_1/L$, $\gamma_{k,2} = \delta_2/L$, and $\delta_1, \delta_2 \in (0,1)$. Assume that $\liminf_{k\to\infty} \lambda_k(2 - \lambda_k) > 0$ and $\liminf_{k\to\infty} \alpha_k > 0$. Let $\kappa = \liminf_{k\to\infty} \lambda_k(2 - \lambda_k)\alpha_k^2$.*

* (i) *If for all $k$, $\delta_1 = \frac{1}{2}, \delta_2 = \frac{1}{2}$ and $\sigma_k = \rho > -\frac{5}{9L}$, then the sequence $(\bar{\boldsymbol{z}}^k)_{k\in\mathbb{N}}$ generated by ($n$-step EG) satisfies $\min_{k=0,1,\ldots,m}\|F\bar{\boldsymbol{z}}^k\|^2 \leq \frac{1}{\kappa(m+1)}\|\boldsymbol{z}^0 - \boldsymbol{z}^*\|^2$.*

* (ii) *If for all $k$, $\delta_1 = \hat{\delta}_1, \delta_2 = \hat{\delta}_2$ and $\sigma_k = \rho > -\frac{\zeta}{L}$, where $\hat{\delta}_1 \approx 0.52212, \hat{\delta}_2 \approx 0.644793$ is the unique solution of following equations,*

$$\begin{cases} \delta_2^2(1 + \delta_1^2) = (1 - \delta_1^2)^2 \\ \delta_1^6 = (1 + \delta_1^2)(1 - \delta_2)^4 \\ \delta_1 + \delta_2 > 1, \delta_1, \delta_2 < 1 \end{cases} \qquad (4.3)$$

  *and $\zeta = \frac{\hat{\delta}_1(1-\hat{\delta}_1^2-\hat{\delta}_2^2)}{2(1-\hat{\delta}_1^2)(1-\hat{\delta}_2^2)} + \frac{\hat{\delta}_2}{1+\hat{\delta}_2} \approx 0.5834$, then the sequence $(\bar{\boldsymbol{z}}^k)_{k\in\mathbb{N}}$ generated by ($n$-step EG) satisfies $\min_{k=0,1,\ldots,m}\|F\bar{\boldsymbol{z}}^k\|^2 \leq \frac{1}{\kappa(m+1)}\|\boldsymbol{z}^0 - \boldsymbol{z}^*\|^2$.*

The theorem presents quite interesting results. When the sum of sub-iteration stepsizes is bounded by $1/L$, a familiar stepsize choice in preliminary algorithms, the optimal range is attained under a succinct invariant stepsize setting. However in otherwise situation, suggested parameters are highly complicated varying stepsizes. Note that invariant and varying both refer to sub-iteration stepsizes $\gamma_{k,1}, \ldots, \gamma_{k,n}$ here and in the subsequent discussion. See Fig. 6 in Appendix C.1 for a contour of (4.2) that incorporates the results in Theorem 4.1 and 4.2.

## 4.2 $n$-STEP EXTRAGRADIENT

While the case of $\sum_{i=1}^{n} \gamma_{k,i} L > 1$ is convoluted even when $n = 2$, we are able to generalize the results when $\sum_{i=1}^{n} \gamma_{k,i} L \leq 1$ to $n \geq 3$ cases:

**Theorem 4.3.** *Let F be L-Lipschitz and satisfy weak Minty condition with $\rho$. Assume that for all $k$ and $i \in [n]$, $\gamma_{k,i} = \delta_i/L$, $\delta_i \in (0,1)$ and $\sum_{i=1}^{n} \delta_i \leq 1$. Assume that $\liminf_{k\to\infty} \lambda_k(2 - \lambda_k) > 0$ and $\liminf_{k\to\infty} \alpha_k > 0$. If for all $k$, $\sigma_k \leq \rho$ and*

$$\sigma_k > -\frac{1}{L}\left(1 - \prod_{i=1}^{n}\frac{1}{1+\delta_i}\right) \tag{4.4}$$

*Then the sequence $(\bar{z}^k)_{k\in\mathbb{N}}$ generated by (n-step EG) satisfies $\min_{k=0,1,\ldots,m}\|F\bar{z}^k\|^2 \leq \frac{1}{\kappa(m+1)}\|z^0 - z^*\|^2$, where $\kappa = \liminf_{k\to\infty}\lambda_k(2 - \lambda_k)\alpha_k^2$.*

Given the positive result extended, this next theorem expands our understanding of parameter selection from point to line, thus broadening the algorithm's adaptability.

**Theorem 4.4.** *Let F be L-Lipschitz and satisfy weak Minty condition with $\rho$. Assume that for all $k$ and $i \in [n]$, $\gamma_{k,i} = \gamma = \delta/L$. Assume that $\liminf_{k\to\infty}\lambda_k(2 - \lambda_k) > 0$ and $\liminf_{k\to\infty}\alpha_k > 0$.*

*(i) If $\delta \in \left(\lfloor \frac{1}{\sqrt[n]{1+\rho L}} - 1\rfloor_+, \frac{1}{n}\right]$, $\sigma_k \in \left(-n\left[1 - \frac{1}{(1+\frac{1}{n})^n}\right]\gamma, \rho\right]$ and $\rho > -\frac{1}{L}\left[1 - \frac{1}{(1+\frac{1}{n})^n}\right]$, where $\lfloor x \rfloor_+ := \max\{0, x\}$, then the sequence $(\bar{z}^k)_{k\in\mathbb{N}}$ generated by (n-step EG) satisfies $\min_{k=0,1,\ldots,m}\|F\bar{z}^k\|^2 \leq \frac{1}{\kappa(m+1)}\|z^0 - z^*\|^2$, where $\kappa = \liminf_{k\to\infty}\lambda_k(2 - \lambda_k)\alpha_k^2$.*

*(ii) Given any $\rho > -\frac{1}{L}(1 - \frac{1}{e})$, let $n \geq \lceil \frac{1}{2+2\log(1+\rho L)}\rceil$, then $\rho > -\frac{1}{L}\left[1 - \frac{1}{(1+\frac{1}{n})^n}\right]$.*

Theorem 4.4 *(i)* provides the ranges for $\gamma$ and $\sigma_k$ under invariant stepsize. When $n = 1$, the algorithm recovers the parameter range $\gamma \in \left(\lfloor-\frac{\rho}{1+\rho L}\rfloor_+, 1/L\right]$ and $\sigma_k \in (-\gamma/2, \rho]$ from (AdaptiveEG+). Selecting $\sigma_k$ near its lower bound is the common practice when $\rho$ is unknown. Theorem 4.4 *(ii)* establishes the global range of $\rho \in (-(1-1/e)/L, \infty)$ in this paper.

We remark that the range of $\rho$ in Theorem 4.4 *(i)* is not the global optimal result. It can be improved if there are no restriction on $\sum_{i=1}^{n}\gamma_{k,i}L$, just as in the case of $n = 2$. Yet it is interesting to figure out whether the global optimum of $-\rho_0 L$ converges to $1 - 1/e$ as $n \to \infty$. Our answer is no. In numerical experiments, 3-step EG with $\delta_1 \approx 0.272899, \delta_2 \approx 0.512753, \delta_3 \approx 0.515522$ demonstrates a range of $-\rho_0 L \approx 0.632242$, which already exceeds $1 - 1/e$, and 4-step EG improves it to at least $-\rho_0 L \approx 0.657724$. Refer to Appendix C.4 for more details. This leaves room for further enhancements.

## 5 ADAPTIVE EXPLORATION BY PURSUING MAX DISTANCE

It is discussed in Section 4 that increasing extrapolation stepsize push the lower bound of $\rho$ towards the threshold of $-1/L$. Inspired by this we propose an algorithm that explores aggressively. In the following scheme, extrapolation process will not stop until projection distance stops increasing.

The algorithm aims to find a projection distance as large as possible. Thanks to the convergence results in Section 4.2, choosing early stepsizes in accordance with (n-step EG) guarantees a positive distance for problems with $\rho > -\frac{1}{L}\left(1 - \frac{1}{e}\right)$ and subsequent explorations will only increase it.

Note that the distances may converge and thus be monotonic, when the GDA sub-iteration of $z_i^k$ directly converges. Therefore a very small tolerance $\varepsilon_1 > 0$ is introduced to preclude the sub-iterations from endless loop. The algorithm may have worse complexity in such scenarios when $F$ are more structured than monotone, reflecting a trade-off between complexity in easier problems and convergence in a broader class. $\varepsilon_2$ recovers GDA when the algorithm potentially stagnates due to intractable local environment, thereby circumvents thorny areas (see Example 3 and Appendix D.3).

**Parameter choice and knowledge of $\rho$** By setting $\sigma_k = \rho$, Algorithm 1 exploit the information of $\rho$ to the largest extent. A smaller $\sigma_k \leq \rho$ still guarantees convergence as long as $\alpha_k > 0$, since

---

**Algorithm 1** Max Distance Extragradient

---

**initialize:**
 $z^0 \in \mathbb{R}^n, \rho, \lambda_k \in (0, 2), \gamma_k \in (0, 1/L], \sigma_k \leq \rho$ and tolerance $\varepsilon_1, \varepsilon_2 > 0$
**repeat** for $k = 0, 1, \ldots$
  Let $z_0^k = z^k, d_0^k = -\infty$
  **repeat** for $i = 1, 2, \ldots$
    Let $z_i^k = z_{i-1}^k - \gamma_k F z_{i-1}^k$
    Compute estimated distance

$$d_i^k = \frac{\sigma_k \|F z_i^k\|^2 - \langle F z_i^k, z_i^k - z^k \rangle}{\|F z_i^k\|}$$

  **until** $d_i^k - d_{i-1}^k < \varepsilon_1 \|F z_i^k\|$
  Let $\bar{z}^k = z_{i-1}^k, \bar{d}^k = d_{i-1}^k$
  Compute stepsize

$$\alpha_k = \frac{\bar{d}^k}{\|F \bar{z}^k\|} = \sigma_k - \frac{\langle F \bar{z}^k, \bar{z}^k - z^k \rangle}{\|F \bar{z}^k\|^2}$$

  Update
  **if** $\alpha_k \geq \varepsilon_2$ **then** $z^{k+1} = z^k - \lambda_k \alpha_k F \bar{z}^k$
  **else** $z^{k+1} = \bar{z}^k$
**until** convergence
**return** $z^{k+1}$

---

projection onto a larger halfspace still proceed toward the goal. A safe parameter range can be found in Theorem 4.4 *(i)*. More aggressive candidates such as $\sigma_k = -1/L$ may apply to harder problems.

## 6 EXAMPLES AND EXPERIMENTS

We consider three classic examples corresponding to $\rho > -\frac{1-\frac{1}{e}}{L}$, $\rho \in \left(-\frac{1}{L}, -\frac{1-\frac{1}{e}}{L}\right)$, $\rho < -\frac{1}{L}$ respectively.

**Example 1.** (bilinear)

$$\min_{x \in \mathbb{R}} \max_{y \in \mathbb{R}} f(x, y) := axy + \frac{b}{2}(x^2 - y^2) \tag{Bilinear}$$

*where $a > 0$, $b < 0$.*

**Example 2.** ((Pethick et al., 2022), Example 3)

$$F z = (\psi(x, y) - y, \psi(y, x) - x) \tag{PolarGame}$$

*where $\psi(x, y) = \frac{1}{16} ax(-1 + x^2 + y^2)(-9 + 16x^2 + y^2)$ and $a = 1$.*

**Example 3.** ((Hsieh et al., 2021), Example 5.2)

$$\min_{x \in \mathbb{R}} \max_{y \in \mathbb{R}} f(x, y) := x(y - 0.45) + \phi(x) - \phi(y) \tag{Forsaken}$$

*where $\phi(z) = \frac{1}{4} z^2 - \frac{1}{2} z^4 + \frac{1}{6} z^6$.*

Tested algorithms include ($n$-step EG), (MDEG) in this paper, (AdaptiveEG+), (CurvatureEG+) from (Pethick et al., 2022) and (EG+ Adaptive) from (Böhm, 2022). All experiments are implemented without the knowledge of $\rho$. In Example 1 we choose the parameters to make $\rho L \in (-0.6, -0.5)$ and verify the convergence result of ($n$-step EG) in Theorem 4.4*(i)*. Example 2 exhibits two limit cycles, one attracting and one repellent. (Algorithm 1) excels at handling such cyclic problems and evades the limit cycle in the first iteration. Example 3 further exceeds the manageable threshold of $\rho > -1/L$, posing challenges for the algorithms. The basic version of (Algorithm 1) erroneously stagnates in the problematic area, echoing the discussion on intractability. Introducing the tolerance $\varepsilon_2$ prevents the algorithms from incorrect convergence and helps to recover GDA when in a predicament. The modified algorithm circumvents the thorny area, as shown in Fig. 4. Among all tested algorithms, (Algorithm 1) and (CurvatureEG+) converge in Example 2 and 3, where our method relies solely on global information.

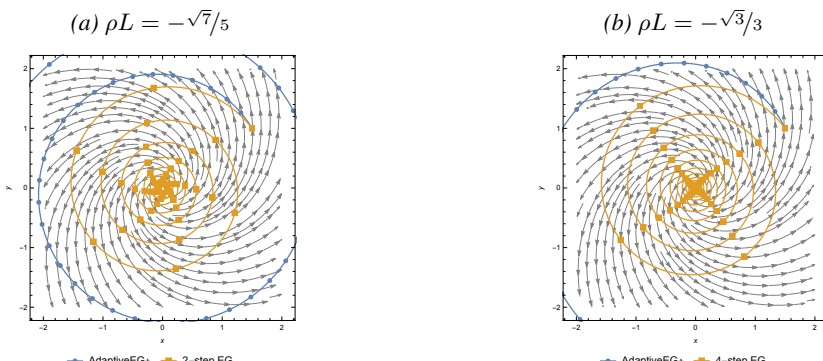

Figure 2: *Example 1.* $\rho < -1/2L$ *examples which are beyond the lower bound of (AdaptiveEG+) and (n-step EG) converges with guarantee.*

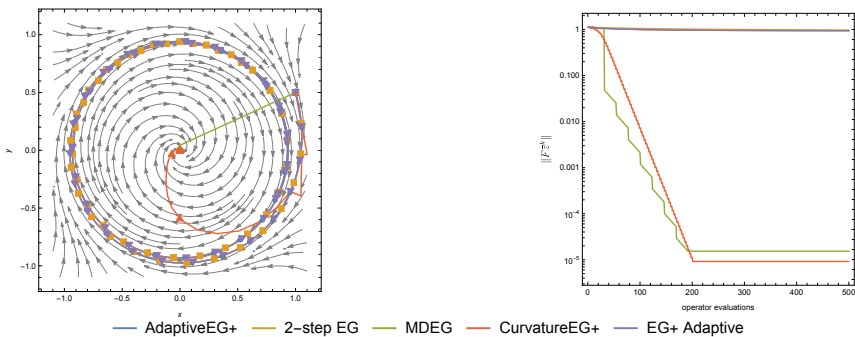

Figure 3: *Example 2 with* $\rho L \approx -0.885521$ *in the box* $\|z\|_\infty \leq 3/2$. *(MDEG) break out of the limit cycle and converges to the stationary point. The first 5 iterations of (MDEG) are attained after 30, 23, 23, 23, 23 exploration steps respectively.*

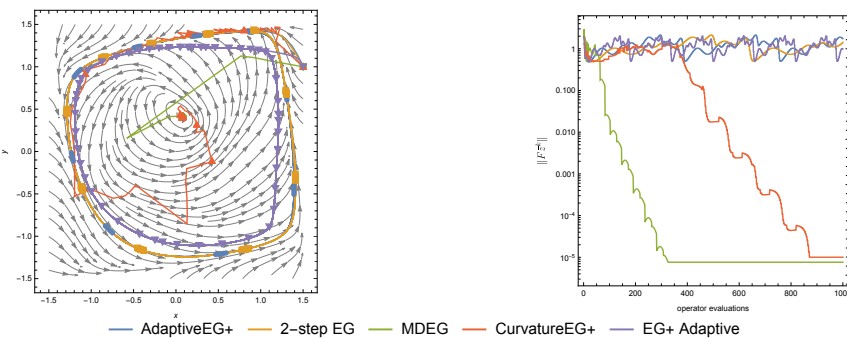

Figure 4: *Example 3 has highly nonmonotonic regions with local* $\rho L \approx -3.04076$ *which go beyond convergence guarantee of all shown algorithms. (MDEG) bypasses the thorny areas and converges to the stationary point, outperforming counterparts on operator evaluations.*

## 7 CONCLUSION

This paper opens up a new dimension for extragradient-type algorithms and demonstrates how expanding extrapolation could help address more problems. We provide bound analysis on our framework of multi-step extrapolation EG+ algorithms, relax the condition $\rho > -1/2L$ to $\rho > -(1-1/e)/L$ and capture past algorithms as special cases. Furthermore, the adaptive method we propose effectively resolves problems with limit cycles. While our method utilizes repeated GDA steps in its subroutine, investigating alternative subroutines that offer better approximation of the proximal point operator could represent a valuable research direction.

ACKNOWLEDGMENTS

This work has been supported by National Natural Science Foundation of China (NSFC) Grant No. 62073294, No. U2341216. We would like to thank the anonymous reviewers for spending time and efforts and bringing in many insightful comments and suggestions, which greatly contributed to the improvement of this work.

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

## A  TECHNICAL LEMMAS

**Lemma A.1.** *Let $F$ be L-Lipschitz and satisfy weak Minty condition with $\rho$. Assume that for any $z^k \in \mathbb{R}^d$, there is $\bar{z}^k \in \mathbb{R}^d$ such that $d(z^k, \mathcal{D}(\bar{z}^k)) > 0$. Let $\sigma_k \le \rho$, $\lambda_k > 0$. Let $\alpha_k = \sigma_k - \frac{\langle F\bar{z}^k, \bar{z}^k - z^k \rangle}{\|F\bar{z}^k\|^2}$, $z^{k+1} = z^k - \lambda_k \alpha_k F\bar{z}^k$. Assume that for all $k$, $\alpha_k > 0$. Then,*

$$\|z^{k+1} - z^*\|^2 \le \|z^k - z^*\|^2 - \lambda_k (2 - \lambda_k)\alpha_k^2 \|F\bar{z}^k\|^2 \tag{A.1}$$

*Proof.* From the expression of $\alpha_k$ we know that

$$\langle F\bar{z}^k, \bar{z}^k - z^k \rangle = (\sigma_k - \alpha_k)\|F\bar{z}^k\|^2 \tag{A.2}$$

$$\le (\rho - \alpha_k)\|F\bar{z}^k\|^2 \tag{A.3}$$

Together with Assumption 2

$$\langle F\bar{z}^k, z^k - z^* \rangle \ge \alpha_k \|F\bar{z}^k\|^2 \tag{A.4}$$

Therefore,

$$\|z^{k+1} - z^*\|^2 = \|z^k - z^* - \lambda_k \alpha_k F\bar{z}^k\|^2 \tag{A.5}$$

$$= \|z^k - z^*\|^2 - 2\lambda_k \alpha_k \langle F\bar{z}^k, z^k - z^* \rangle + \lambda_k^2 \alpha_k^2 \|F\bar{z}^k\|^2 \tag{A.6}$$

$$\le \|z^k - z^*\|^2 - 2\lambda_k \alpha_k^2 \|F\bar{z}^k\|^2 + \lambda_k^2 \alpha_k^2 \|F\bar{z}^k\|^2 \tag{A.7}$$

$$= \|z^k - z^*\|^2 - \lambda_k (2 - \lambda_k)\alpha_k^2 \|F\bar{z}^k\|^2 \tag{A.8}$$

$\square$

**Remark 1.** It is worth mentioning that Lemma A.1 is highly related to (Solodov & Svaiter, 1999), Lemma 2.1.

Notice that when $\lambda_k = 1$,

$$\langle F\bar{z}^k, z^k - \bar{z}^k + \sigma_k F\bar{z}^k \rangle = \alpha_k \|F\bar{z}^k\|^2 > 0 \tag{A.9}$$

$$\langle F\bar{z}^k, z^* - \bar{z}^k + \sigma_k F\bar{z}^k \rangle = (\sigma_k - \rho)\|F\bar{z}^k\|^2 \le 0 \tag{A.10}$$

and $z^{k+1} = P_{\mathcal{H}}[z^k] = z^k - \alpha_k F\bar{z}^k$ where $P_{\mathcal{H}}[z^k]$ is the projection of $z^k$ onto the hyperplane $\mathcal{H} = \{w \in \mathbb{R}^d \mid \langle F\bar{z}^k, w - \bar{z}^k + \sigma_k F\bar{z}^k \rangle = 0\}$, which is evident from (A.9).

According to (Solodov & Svaiter, 1999), Lemma 2.1,

$$\|z^k - z^*\|^2 \ge \|z^{k+1} - z^*\|^2 + \|z^{k+1} - z^k\|^2 \tag{A.11}$$

$$= \|z^{k+1} - z^*\|^2 + \alpha_k^2 \|F\bar{z}^k\|^2 \tag{A.12}$$

which is in accordance with Lemma A.1.

**Lemma A.2.** *Let $F$ be L-Lipschitz and satisfy weak Minty condition with $\rho$. Let $n = 2$. Assume that for all $k$, $\gamma_{k,1} = \delta_1/L$, $\gamma_{k,2} = \delta_2/L$, and $\delta_1, \delta_2 \in (0, 1)$. Define function $g : (0, 1) \times (0, 1) \to \mathbb{R}$ as*

$$g(\delta_1, \delta_2) := \begin{cases} \frac{1}{(1+\delta_1)(1+\delta_2)} & \text{if } \delta_1 + \delta_2 \le 1 \\ \frac{1 - \delta_1^2 - \delta_2^2}{2(1 - \delta_1^2)(1 - \delta_2^2)} & \text{if } \delta_1 + \delta_2 > 1 \end{cases} \tag{A.13}$$

*Initialize $z^k$ and generate the sequence $(z_i^k)_{i \in [n]}$ by (n-step EG). The following inequality holds and is tight:*

$$\frac{\langle Fz_2^k, Fz^k \rangle}{\|Fz_2^k\|^2} \ge g(\delta_1, \delta_2) \tag{A.14}$$

*Proof.* In this proof we treat $Fz_2^k$ as an axis. Define following as coordinates with respect to $Fz_2^k$,

$$x := \frac{\langle Fz^k, Fz_2^k \rangle}{\|Fz_2\|^2} \tag{A.15}$$

$$x_1 := \frac{\langle Fz_1^k, Fz_2^k \rangle}{\|Fz_2\|^2} \tag{A.16}$$

From ($n$-step EG) and Lipschitz continuity (Assumption 1), it is clear that

$$\|Fz^k - Fz_1^k\| \le \gamma_{k,1} L\|Fz^k\| = \delta_1\|Fz^k\| \tag{A.17}$$

$$\|Fz_1^k - Fz_2^k\| \le \gamma_{k,2} L\|Fz_1^k\| = \delta_2\|Fz_1^k\| \tag{A.18}$$

Changing the perspective to fixed $Fz_1^k$ and $Fz_2^k$ we expand and recomplete the square

$$\|Fz^k - \frac{1}{1-\delta_1^2}Fz_1^k\| \le \frac{\delta_1}{1-\delta_1^2}\|Fz_1^k\| \tag{A.19}$$

$$\|Fz_1^k - \frac{1}{1-\delta_2^2}Fz_2^k\| \le \frac{\delta_2}{1-\delta_2^2}\|Fz_2^k\| \tag{A.20}$$

where $Fz_1^k$ is in a ball whose center and radius is proportional to $Fz_2^k$, and $Fz^k$ is in a ball whose center and radius is proportional to $Fz_1^k$. The boundary of such area is called a Cartesian oval in two dimension, or a Cartesian surface in three or more dimensions.

Apply Cauchy-Schwarz inequality on (A.20)

$$-\frac{\delta_2}{1-\delta_2^2}\|Fz_2^k\|^2 \le \langle Fz_1^k - \frac{1}{1-\delta_2^2}Fz_2^k, Fz_2^k \rangle \le \frac{\delta_2}{1-\delta_2^2}\|Fz_2^k\|^2 \tag{A.21}$$

$$\frac{1}{1+\delta_2}\|Fz_2^k\|^2 \le \langle Fz_1^k, Fz_2^k \rangle \le \frac{1}{1-\delta_2}\|Fz_2^k\|^2 \tag{A.22}$$

$$\frac{1}{1+\delta_2} \le x_1 \le \frac{1}{1-\delta_2} \tag{A.23}$$

Apply Cauchy-Schwarz inequality on (A.19)

$$\langle Fz^k - \frac{1}{1-\delta_1^2}Fz_1^k, Fz_2^k \rangle \ge -\|Fz^k - \frac{1}{1-\delta_1^2}Fz_1^k\|\|Fz_2^k\| \tag{A.24}$$

$$\ge -\frac{\delta_1}{1-\delta_1^2}\|Fz_1^k\|\|Fz_2^k\| \tag{A.25}$$

$$\langle Fz^k, Fz_2^k \rangle \ge \frac{1}{1-\delta_1^2}\langle Fz_1^k, Fz_2^k \rangle - \frac{\delta_1}{1-\delta_1^2}\|Fz_1^k\|\|Fz_2^k\| \tag{A.26}$$

Divide both sides with $\|Fz_2^k\|^2$ we have

$$x \ge \frac{1}{1-\delta_1^2}x_1 - \frac{\delta_1}{1-\delta_1^2}\frac{\|Fz_1^k\|}{\|Fz_2^k\|} \tag{A.27}$$

From $\|Fz_1^k - Fz_2^k\| \le \delta_2\|Fz_1^k\|$ it is straightforward to see $\|Fz_1^k\|^2 \le \frac{2}{1-\delta_2^2}\langle Fz_1^k, Fz_2^k \rangle - \frac{1}{1-\delta_2^2}\|Fz_2^k\|^2$, deriving that $\frac{\|Fz_1^k\|}{\|Fz_2^k\|} \le \sqrt{\frac{2x_1-1}{1-\delta_2^2}}$. Therefore,

$$x \ge \frac{1}{1-\delta_1^2}\left(x_1 - \delta_1\sqrt{\frac{2x_1-1}{1-\delta_2^2}}\right) \tag{A.28}$$

Define function $p : [\frac{1}{1+\delta_2}, \frac{1}{1-\delta_2}] \to \mathbb{R}$ as $p(x_1) := \frac{1}{1-\delta_1^2}\left(x_1 - \delta_1\sqrt{\frac{2x_1-1}{1-\delta_2^2}}\right)$

$$p'(x_1) = \frac{1}{1-\delta_1^2}\left(1 - \frac{\delta_1}{\sqrt{(1-\delta_2^2)(2x_1-1)}}\right) \tag{A.29}$$

which is monotonically increasing. Let $p'(x_1) = 0$, the solution is $x_1 = \frac{1+\delta_1^2-\delta_2^2}{2(1-\delta_2^2)}$. Whether this extremum point fall into the domain hinge upon the relation between $\frac{1+\delta_1^2-\delta_2^2}{2(1-\delta_2^2)}$ and $\frac{1}{1+\delta_2}$, since it is easy to examine that $\frac{1+\delta_1^2-\delta_2^2}{2(1-\delta_2^2)} < \frac{1}{1-\delta_2}$.

If $\frac{1+\delta_1^2-\delta_2^2}{2(1-\delta_2^2)} \leq \frac{1}{1+\delta_2}$, equivalently $\delta_1^2 \leq (1-\delta_2)^2$, $\delta_1 + \delta_2 \leq 1$

$$x \geq p(\frac{1}{1+\delta_2}) = \frac{1}{(1+\delta_1)(1+\delta_2)} \tag{A.30}$$

If $\frac{1+\delta_1^2-\delta_2^2}{2(1-\delta_2^2)} > \frac{1}{1+\delta_2}$, equivalently $\delta_1^2 > (1-\delta_2)^2$, $\delta_1 + \delta_2 > 1$

$$x \geq p(\frac{1+\delta_1^2-\delta_2^2}{2(1-\delta_2^2)}) = \frac{1-\delta_1^2-\delta_2^2}{2(1-\delta_1^2)(1-\delta_2^2)} \tag{A.31}$$

establishing the lemma. $\qquad\square$

**Lemma A.3.** *Let F be L-Lipschitz and satisfy weak Minty condition with $\rho$. Assume that for all $k$, $\gamma_{k,i} = \delta_i/L$, $\delta_i \in (0,1)$, $i=1,\ldots,n$ and $\sum_{i=1}^n \delta_i \leq 1$. Initialize $\mathbf{z}^k$ and generate the sequence $(\mathbf{z}_i^k)_{i \in [n]}$ by (n-step EG). The following inequality holds and is tight:*

$$\frac{\langle F\mathbf{z}_n^k, F\mathbf{z}^k \rangle}{\|F\mathbf{z}_n^k\|^2} \geq \prod_{i=1}^n \frac{1}{1+\delta_i} \tag{A.32}$$

*Proof.* Similar to Lemma A.2, what we want to prove is that the minimum value is attained when $F\mathbf{z}^k$ is scalar multiple of $F\mathbf{z}_n^k$. We make the following key proposition and prove it using mathematical induction:

$$\|F\mathbf{z}^k - \frac{1}{(1-\sum_{i=1}^n \delta_i)\prod_{i=1}^n(1+\delta_i)}F\mathbf{z}_n^k\| \leq \frac{\sum_{i=1}^n \delta_i}{(1-\sum_{i=1}^n \delta_i)\prod_{i=1}^n(1+\delta_i)}\|F\mathbf{z}_n^k\| \tag{A.33}$$

For $n=1$, it is mentioned in (A.19) that $\|F\mathbf{z}^k - \frac{1}{1-\delta_1^2}F\mathbf{z}_1^k\| \leq \frac{\delta_1}{1-\delta_1^2}\|F\mathbf{z}_1^k\|$.

Assume that the proposition holds for $n = m-1$. Apply the result on the $m-1$ extrapolations from $\mathbf{z}_1^k$ to $\mathbf{z}_m^k$,

$$\|F\mathbf{z}_1^k - \frac{1}{(1-\sum_{i=2}^m \delta_i)\prod_{i=2}^m(1+\delta_i)}F\mathbf{z}_m^k\| \leq \frac{\sum_{i=2}^m \delta_i}{(1-\sum_{i=2}^m \delta_i)\prod_{i=2}^m(1+\delta_i)}\|F\mathbf{z}_m^k\| \tag{A.34}$$

Square both sides and rearrange the equation,

$$\langle F\mathbf{z}_1^k, F\mathbf{z}_m^k \rangle \geq \frac{(1-\sum_{i=2}^m \delta_i)\prod_{i=2}^m(1+\delta_i)}{2}\|F\mathbf{z}_1^k\|^2 + \frac{1+\sum_{i=2}^m \delta_i}{2\prod_{i=2}^m(1+\delta_i)}\|F\mathbf{z}_m^k\|^2 \tag{A.35}$$

To prove the $n = m$ occasion, we examine the correctness of following inequality.

$$\|\frac{1}{1-\delta_1^2}F\mathbf{z}_1^k - \frac{1}{(1-\sum_{i=1}^m \delta_i)\prod_{i=1}^m(1+\delta_i)}F\mathbf{z}_m^k\| \leq \frac{\sum_{i=1}^m \delta_i}{(1-\sum_{i=1}^m \delta_i)\prod_{i=1}^m(1+\delta_i)}\|F\mathbf{z}_m^k\| - \frac{\delta_1}{1-\delta_1^2}\|F\mathbf{z}_1^k\|$$

$$\tag{A.36}$$

$$\|\frac{1}{1-\delta_1^2}F\boldsymbol{z}_1^k - \frac{1}{(1-\sum\limits_{i=1}^{m}\delta_i)\prod\limits_{i=1}^{m}(1+\delta_i)}F\boldsymbol{z}_m^k\|^2 - \left(\frac{\sum\limits_{i=1}^{m}\delta_i}{(1-\sum\limits_{i=1}^{m}\delta_i)\prod\limits_{i=1}^{m}(1+\delta_i)}\|F\boldsymbol{z}_m^k\| - \frac{\delta_1}{1-\delta_1^2}\|F\boldsymbol{z}_1^k\|\right)^2$$

$$=\frac{1}{1-\delta_1^2}\|F\boldsymbol{z}_1^k\|^2 + \frac{1+\sum\limits_{i=1}^{m}\delta_i}{(1-\sum\limits_{i=1}^{m}\delta_i)\prod\limits_{i=1}^{m}(1+\delta_i)^2}\|F\boldsymbol{z}_m^k\|^2 + \frac{2\delta_1\sum\limits_{i=1}^{m}\delta_i}{(1-\delta_1^2)(1-\sum\limits_{i=1}^{m}\delta_i)\prod\limits_{i=1}^{m}(1+\delta_i)}\|F\boldsymbol{z}_1^k\|\|F\boldsymbol{z}_m^k\|$$

$$-\frac{2}{(1-\delta_1^2)(1-\sum\limits_{i=1}^{m}\delta_i)\prod\limits_{i=1}^{m}(1+\delta_i)}\langle F\boldsymbol{z}_1^k, F\boldsymbol{z}_m^k\rangle$$

$$\leq\frac{1}{1-\delta_1^2}\|F\boldsymbol{z}_1^k\|^2 + \frac{1+\sum\limits_{i=1}^{m}\delta_i}{(1-\sum\limits_{i=1}^{m}\delta_i)\prod\limits_{i=1}^{m}(1+\delta_i)^2}\|F\boldsymbol{z}_m^k\|^2 + \frac{2\delta_1\sum\limits_{i=1}^{m}\delta_i}{(1-\delta_1^2)(1-\sum\limits_{i=1}^{m}\delta_i)\prod\limits_{i=1}^{m}(1+\delta_i)}\|F\boldsymbol{z}_1^k\|\|F\boldsymbol{z}_m^k\|$$

$$-\frac{1-\sum\limits_{i=2}^{m}\delta_i}{(1-\delta_1^2)(1-\sum\limits_{i=1}^{m}\delta_i)(1+\delta_1)}\|F\boldsymbol{z}_1^k\|^2 - \frac{1+\sum\limits_{i=2}^{m}\delta_i}{(1-\delta_1^2)(1-\sum\limits_{i=1}^{m}\delta_i)\prod\limits_{i=1}^{m}(1+\delta_i)\prod\limits_{i=2}^{m}(1+\delta_i)}\|F\boldsymbol{z}_m^k\|^2$$

$$=-\frac{\delta_1\sum\limits_{i=1}^{m}\delta_i}{(1-\delta_1^2)(1-\sum\limits_{i=1}^{m}\delta_i)(1+\delta_1)}\|F\boldsymbol{z}_1^k\|^2 - \frac{\delta_1(1+\delta_1)\sum\limits_{i=1}^{m}\delta_i}{(1-\delta_1^2)(1-\sum\limits_{i=1}^{m}\delta_i)\prod\limits_{i=1}^{m}(1+\delta_i)^2}\|F\boldsymbol{z}_m^k\|^2$$

$$+\frac{2\delta_1\sum\limits_{i=1}^{m}\delta_i}{(1-\delta_1^2)(1-\sum\limits_{i=1}^{m}\delta_i)\prod\limits_{i=1}^{m}(1+\delta_i)}\|F\boldsymbol{z}_1^k\|\|F\boldsymbol{z}_m^k\|$$

$$=-\frac{\delta_1\sum\limits_{i=1}^{m}\delta_i}{(1-\delta_1^2)(1-\sum\limits_{i=1}^{m}\delta_i)(1+\delta_1)}\left(\|F\boldsymbol{z}_1^k\|^2 + \frac{1}{\prod\limits_{i=2}^{m}(1+\delta_i)^2}\|F\boldsymbol{z}_m^k\|^2 - \frac{2}{\prod\limits_{i=2}^{m}(1+\delta_i)}\|F\boldsymbol{z}_1^k\|\|F\boldsymbol{z}_m^k\|\right)$$

$$=-\frac{\delta_1\sum\limits_{i=1}^{m}\delta_i}{(1-\delta_1^2)(1-\sum\limits_{i=1}^{m}\delta_i)(1+\delta_1)}\left(\|F\boldsymbol{z}_1^k\| - \frac{1}{\prod\limits_{i=2}^{m}(1+\delta_i)}\|F\boldsymbol{z}_m^k\|\right)^2 \leq 0$$

$$\tag{A.37}$$

It is rather easy to examine the positiveness of the right-hand side

$$\frac{\sum\limits_{i=1}^{m}\delta_i}{(1-\sum\limits_{i=1}^{m}\delta_i)\prod\limits_{i=1}^{m}(1+\delta_i)}\|F\boldsymbol{z}_m^k\| \geq \frac{\sum\limits_{i=1}^{m}\delta_i}{(1-\sum\limits_{i=1}^{m}\delta_i)\prod\limits_{i=1}^{m-1}(1+\delta_i)}\|F\boldsymbol{z}_{m-1}^k\|$$

$$\vdots$$

$$\geq \frac{\sum\limits_{i=1}^{m}\delta_i}{(1-\sum\limits_{i=1}^{m}\delta_i)(1+\delta_1)}\|F\boldsymbol{z}_1^k\|$$

$$\geq \frac{\delta_1}{(1-\delta_1)(1+\delta_1)}\|F\boldsymbol{z}_1^k\| = \frac{\delta_1}{1-\delta_1^2}\|F\boldsymbol{z}_1^k\|$$

$$\tag{A.38}$$

The above two formulae complete the proof of (A.36), and the $n = m$ case follows from triangle inequality and $\|F\boldsymbol{z}^k - \frac{1}{1-\delta_1^2}F\boldsymbol{z}_1^k\| \leq \frac{\delta_1}{1-\delta_1^2}\|F\boldsymbol{z}_1^k\|$. The proposition (A.33) is then proved by induction. Using Cauchy-Schwarz inequality we obtain

$$\langle F\boldsymbol{z}_n^k, F\boldsymbol{z}^k - \frac{1}{(1 - \sum\limits_{i=1}^{n} \delta_i) \prod\limits_{i=1}^{n} (1 + \delta_i)} F\boldsymbol{z}_n^k \rangle \geq -\frac{\sum\limits_{i=1}^{n} \delta_i}{(1 - \sum\limits_{i=1}^{n} \delta_i) \prod\limits_{i=1}^{n} (1 + \delta_i)} \|F\boldsymbol{z}_n^k\|^2 \tag{A.39}$$

$$\langle F\boldsymbol{z}_n^k, F\boldsymbol{z}^k \rangle \geq \frac{1}{\prod\limits_{i=1}^{n} (1 + \delta_i)} \|F\boldsymbol{z}_n^k\|^2 \tag{A.40}$$

establishing the lemma. $\qquad\square$

## B  PROOFS

***Proof of Lemma 2.3.*** Let $\boldsymbol{w}$ be any point on the hyperplane $\partial\mathcal{D}(\boldsymbol{u})$. According to the definition,

$$\langle F\boldsymbol{u}, \boldsymbol{u} - \boldsymbol{w} \rangle = \rho\|F\boldsymbol{u}\|^2 \tag{B.1}$$

$F\boldsymbol{u}$ is perpendicular to the hyperplane, and the distance from $\boldsymbol{v}$ to $\partial\mathcal{D}(\boldsymbol{u})$ is equal to the length of the orthogonal projection of $\boldsymbol{v} - \boldsymbol{w}$ on $F\boldsymbol{u}$.

$$d(\boldsymbol{v}, \partial\mathcal{D}(\boldsymbol{u})) = \|P_{F\boldsymbol{u}}(\boldsymbol{v} - \boldsymbol{w})\| \tag{B.2}$$

$$= \frac{|\langle F\boldsymbol{u}, \boldsymbol{v} - \boldsymbol{w} \rangle|}{\|F\boldsymbol{u}\|} \tag{B.3}$$

$$= \frac{|\langle F\boldsymbol{u}, \boldsymbol{u} - \boldsymbol{w} \rangle - \langle F\boldsymbol{u}, \boldsymbol{u} - \boldsymbol{v} \rangle|}{\|F\boldsymbol{u}\|} \tag{B.4}$$

$$= \frac{|\rho\|F\boldsymbol{u}\|^2 - \langle F\boldsymbol{u}, \boldsymbol{u} - \boldsymbol{v} \rangle|}{\|F\boldsymbol{u}\|} \tag{B.5}$$

To convert this into a signed distance, we remove the absolute value according to Definition 2. If $\boldsymbol{u} \in \mathcal{S}$, $d(\boldsymbol{v}, \partial\mathcal{D}(\boldsymbol{u})) \leq 0$; If $\boldsymbol{u} \in \mathcal{S}^c$, $d(\boldsymbol{v}, \partial\mathcal{D}(\boldsymbol{u})) > 0$. Thus,

$$d(\boldsymbol{v}, \mathcal{D}(\boldsymbol{u})) = \frac{\rho\|F\boldsymbol{u}\|^2 - \langle F\boldsymbol{u}, \boldsymbol{u} - \boldsymbol{v} \rangle}{\|F\boldsymbol{u}\|} \tag{B.6}$$

$\qquad\square$

***Proof of Theorem 3.1.*** Telescoping (A.1) from $k = 0$ to $k = m$,

$$\|\boldsymbol{z}^0 - \boldsymbol{z}^*\|^2 - \|\boldsymbol{z}^{m+1} - \boldsymbol{z}^*\|^2 \geq \sum_{k=1}^{m} \lambda_k(2 - \lambda_k)\alpha_k^2\|F\bar{\boldsymbol{z}}^k\|^2 \tag{B.7}$$

Let $\varepsilon_k := \lambda_k(2 - \lambda_k)\alpha_k^2$ and $\kappa = \liminf_{k\to\infty} \varepsilon_k$

$$\|\boldsymbol{z}^0 - \boldsymbol{z}^*\|^2 \geq \sum_{k=1}^{m} \varepsilon_k\|F\bar{\boldsymbol{z}}^k\|^2 \geq \kappa \sum_{k=1}^{m} \|F\bar{\boldsymbol{z}}^k\|^2 \tag{B.8}$$

Therefore,

$$\min_{k=0,1,\dots,m} \|F\bar{\boldsymbol{z}}^k\|^2 \leq \frac{1}{m+1} \sum_{k=1}^{m} \|F\bar{\boldsymbol{z}}^k\|^2 \leq \frac{1}{\kappa(m+1)} \|\boldsymbol{z}^0 - \boldsymbol{z}^*\|^2 \tag{B.9}$$

Since $\kappa = \liminf_{k\to\infty} \varepsilon_k > 0$, $\{\|F\bar{\boldsymbol{z}}^k\|^2\}_{k\in\mathbb{N}}$ converges to zero. Combined with Lipschitzness, $\{\|\bar{\boldsymbol{z}}^k - \boldsymbol{z}^*\|\}_{k\in\mathbb{N}}$ converges to zero and $\{\bar{\boldsymbol{z}}^k\}_{k\in\mathbb{N}}$ converges to $\boldsymbol{z}^*$. $\qquad\square$

**Proof of Theorem 4.1.** Expand the formula for calculating the stepsize $\alpha_k$,

$$\alpha_k = \sigma_k - \frac{\langle F\boldsymbol{z}_2^k, \boldsymbol{z}_2^k - \boldsymbol{z}^k \rangle}{\|F\boldsymbol{z}_2^k\|^2} = \sigma_k + \frac{\gamma_{k,1}\langle F\boldsymbol{z}_2^k, F\boldsymbol{z}^k \rangle}{\|F\boldsymbol{z}_2^k\|^2} + \frac{\gamma_{k,2}\langle F\boldsymbol{z}_2^k, F\boldsymbol{z}_1^k \rangle}{\|F\boldsymbol{z}_2^k\|^2} \tag{B.10}$$

The lower bounds of the last two terms are established in Lemma A.2 and (A.22),

$$\alpha_k \geq \sigma_k + \gamma_{k,1}g(\delta_1, \delta_2) + \frac{\gamma_{k,2}}{1 + \delta_2}$$

$$= \sigma_k + \frac{1}{L}\left[\delta_1 g(\delta_1, \delta_2) + \frac{\delta_2}{1 + \delta_2}\right] \tag{B.11}$$

Hence, if (4.2) holds, $\alpha_k > 0$. It is also straightforward that $d(\boldsymbol{z}^k, \mathcal{D}(\bar{\boldsymbol{z}}^k)) > 0$ since $\rho \geq \sigma_k$. With the assumptions met, we can refer to Theorem 3.1 for the convergence result. □

**Proof of Theorem 4.2.** *(i)* When $\delta_1 + \delta_2 \leq 1$,

$$\rho \geq \sigma_k > -\frac{1}{L}\left[1 - \frac{1}{(1 + \delta_1)(1 + \delta_2)}\right] \tag{B.12}$$

$$\geq -\frac{1}{L}\left[1 - (\frac{2}{2 + \delta_1 + \delta_2})^2\right] \tag{B.13}$$

$$= -\frac{5}{9L} \tag{B.14}$$

The equality in (B.13) holds when $\delta_1 = \delta_2 = 1/2$.

*(ii)* Define function $q : \{(x, y) \mid x + y > 1, x < 1, y < 1\} \to \mathbb{R}$ as

$$q(\delta_1, \delta_2) := \frac{\delta_1(1 - \delta_1^2 - \delta_2^2)}{2(1 - \delta_1^2)(1 - \delta_2^2)} + \frac{\delta_2}{1 + \delta_2} \tag{B.15}$$

Try to find its critical point,

$$\frac{\partial q}{\partial \delta_1} = \frac{(1 - \delta_1^2)^2 - \delta_2^2(1 + \delta_1^2)}{2(1 - \delta_1^2)^2(1 - \delta_2^2)} = 0 \tag{B.16}$$

$$\frac{\partial q}{\partial \delta_2} = \frac{1}{(1 + \delta_2)^2} - \frac{\delta_1^3 \delta_2}{(1 - \delta_1^2)(1 - \delta_2^2)^2} = 0 \tag{B.17}$$

Equivalently,

$$\delta_2^2(1 + \delta_1^2) = (1 - \delta_1^2)^2 \tag{B.18}$$
$$\delta_1^3 \delta_2(1 + \delta_2)^2 = (1 - \delta_1^2)(1 - \delta_2^2)^2 \tag{B.19}$$

(4.3) follows by rearranging the equations,

$$\delta_1^6 \delta_2^2(1 + \delta_2)^4 = (1 - \delta_1^2)^2(1 - \delta_2^2)^4 \tag{B.20}$$

$$= \delta_2^2(1 + \delta_1^2)(1 - \delta_2^2)^4 \tag{B.21}$$

$$\delta_1^6(1 + \delta_2)^4 = (1 + \delta_1^2)(1 - \delta_2)^4 \tag{B.22}$$

We compute the solution in *Mathematica*,

$$\delta_2 = \left(z; 1 - 13z + 24z^2 - 20z^3 + 16z^4\right)_2^{-1} \tag{B.23}$$

$$\delta_1 = \sqrt{\frac{-540\delta_2^3 + 432\delta_2^2 - 351\delta_2 + 243}{189}} \tag{B.24}$$

where $\delta_2 = \left(z; 1 - 13z + 24z^2 - 20z^3 + 16z^4\right)_2^{-1}$ is the adopted notation for the second root of the polynomial $1 - 13z + 24z^2 - 20z^3 + 16z^4$ in *Mathematica*'s ordering, which is the larger one of its 2 real roots.

The closed form solution of $\delta_2$ can be solved from the quadratic equation:

$$\delta_2 = \frac{5}{16} + \frac{1}{16}\sqrt{-39 + \frac{8}{3^{\frac{2}{3}}}(576 + 7\sqrt{6771})^{\frac{1}{3}} - \frac{8}{3(576 + 7\sqrt{6771})^{\frac{1}{3}}}}$$

$$+ \frac{1}{2}\left[-\frac{39}{32} - \frac{(576 + 7\sqrt{6771})^{\frac{1}{3}}}{8 \cdot 3^{\frac{2}{3}}} + \frac{1}{8(3(576 + 7\sqrt{6771}))^{\frac{1}{3}}}\right. \tag{B.25}$$

$$\left.+ \frac{61}{32\sqrt{-39 + \frac{8}{3^{\frac{2}{3}}}(576 + 7\sqrt{6771})^{\frac{1}{3}} - \frac{8}{3(576 + 7\sqrt{6771})^{\frac{1}{3}}}}}\right]^{\frac{1}{2}}$$

$\square$

***Proof of Theorem 4.3.*** Notice that Lemma A.3 can be applied on parts of the steps,

$$\frac{\langle Fz_n^k, Fz_{n-1}^k\rangle}{\|Fz_n^k\|^2} \geq \frac{1}{1 + \delta_n} \tag{B.26}$$

$$\frac{\langle Fz_n^k, Fz_{n-2}^k\rangle}{\|Fz_n^k\|^2} \geq \frac{1}{(1 + \delta_{n-1})(1 + \delta_n)} \tag{B.27}$$

$$\vdots \tag{B.28}$$

$$\frac{\langle Fz_n^k, Fz^k\rangle}{\|Fz_n^k\|^2} \geq \frac{1}{(1 + \delta_1)\dots(1 + \delta_i)} \tag{B.29}$$

Expand $\alpha_k$,

$$\alpha_k = \sigma_k - \frac{\langle Fz_n^k, z_n^k - z^k\rangle}{\|Fz_n^k\|^2} \tag{B.30}$$

$$= \sigma_k + \frac{\gamma_1\langle Fz_n^k, Fz^k\rangle}{\|Fz_n^k\|^2} + \frac{\gamma_2\langle Fz_n^k, Fz_1^k\rangle}{\|Fz_n^k\|^2} + \dots + \frac{\gamma_n\langle Fz_n^k, Fz_{n-1}^k\rangle}{\|Fz_n^k\|^2} \tag{B.31}$$

$$\geq \sigma_k + \frac{1}{L}\left[\frac{\delta_1}{(1 + \delta_1)\dots(1 + \delta_n)} + \dots + \frac{\delta_{n-1}}{(1 + \delta_{n-1})(1 + \delta_n)} + \frac{\delta_n}{1 + \delta_n}\right] \tag{B.32}$$

$$= \sigma_k + \frac{1}{L}\left[1 - \frac{1}{(1 + \delta_1)\dots(1 + \delta_n)}\right] \tag{B.33}$$

The inequality is tight since equality holds when $Fz^k = \frac{1}{1+\delta_1}Fz_1^k = \dots = \frac{1}{(1+\delta_1)\dots(1+\delta_n)}Fz_n^k$. Thus, the sufficient and necessary condition of $\alpha_k > 0$ is:

$$\sigma_k > -\frac{1}{L}\left[1 - \frac{1}{(1 + \delta_1)\dots(1 + \delta_n)}\right] \tag{B.34}$$

Similarly, the convergence result follows from Theorem 3.1. $\square$

***Proof of Theorem 4.4.*** *(i)* According to AM-GM inequality,

$$(1 + \delta_1)\dots(1 + \delta_n) \leq (1 + \frac{1}{n}\sum_{k=1}^{n}\delta_k)^n \leq (1 + \frac{1}{n})^n \tag{B.35}$$

The equality holds when $\delta_1 = \dots = \delta_n = \frac{1}{n}$. Therefore,

$$\rho > -\frac{1}{L}\left[1 - \frac{1}{(1 + \delta_1)\dots(1 + \delta_n)}\right] \geq -\frac{1}{L}\left[1 - \frac{1}{(1 + \frac{1}{n})^n}\right] \tag{B.36}$$

$\delta \leq \frac{1}{n}$ and $\sigma_k \leq \rho$ directly come from the theorem assumption.

$\delta > \lfloor \frac{1}{\sqrt[n]{1+\rho L}} - 1 \rfloor_+$ follows from $\rho \geq \sigma_k > -\frac{1}{L}\left[1 - \frac{1}{(1+\delta)^n}\right]$.

Moreover,

$$\sigma_k > -\frac{1}{L}\left[1 - \frac{1}{(1+\delta)^n}\right] \geq -\frac{1}{L}\left[1 - \frac{1}{(1+\frac{1}{n})^n}\right] \geq -n\left[1 - \frac{1}{(1+\frac{1}{n})^n}\right]\gamma \tag{B.37}$$

establishing the parameter ranges in Theorem 4.4 *(i)*.

*(ii)* Given $\rho > -\frac{1}{L}(1 - \frac{1}{e})$, let $n = \lceil \frac{1}{2+2\log(1+\rho L)} \rceil$

$$n\log(1+\frac{1}{n}) > n(\frac{1}{n} - \frac{1}{2n^2}) = 1 - \frac{1}{2n} \tag{B.38}$$

$$> 1 - [1 + \log(1+\rho L)] = -\log(1+\rho L) \tag{B.39}$$

$$(1+\frac{1}{n})^n > \frac{1}{1+\rho L} \tag{B.40}$$

$$\rho > -\frac{1}{L}[1 - \frac{1}{(1+\frac{1}{n})^n}] \tag{B.41}$$

establishing Theorem 4.4 *(ii)*. □

## C  FIGURES AND INTUITIONS

### C.1  ADDITIONAL FIGURES

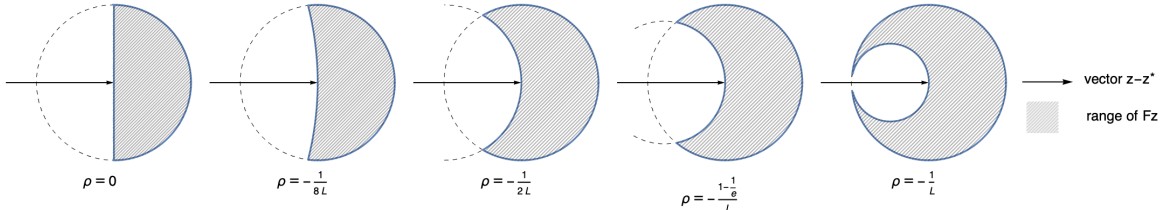

*Figure 5: This figure illustrates the permissible range of the vector $Fz$ under Assumption 2.1 and 2.2. Originating from the circle's center $z$, the vector extends into the shaded area, which is delineated by these assumptions. Notably, the extent of the allowable region is influenced by the relationship between the weak Minty parameter $\rho$ and the Lipschitz constant $L$. Enumerated are several representative problem settings.*

The presented problem settings in Fig. 5 appear in various classic algorithms. $\rho = 0$ recovers Minty variational inequality, also known as star-monotonicity. The MVI ensures that the negative gradient does not point outward from the solution $z^*$. (EG+) extends from (EG) and allows for a slight extent of non-monotinicity with $\rho > -1/8L$. (AdaptiveEG+) further relaxes the problem parameter to $\rho > -1/2L$. These efforts make it permissible for the direction of gradient descent to move away from the solution to a limited extent.

Our methods introduced in the main paper provide new convergence guarantee for problems under $\rho > -(1-1/e)/L$. Furthermore, $\rho > -1/L$ marks the boundary of tractability. As discussed in Section 4, increasing the exploration stepsize could potentially solve more problems within this range.

Fig. 6 visually encapsulates the findings regarding 2-step EG in Section 4.1. The plot is a contour of the following function, in the box $[0, 0.8] \times [0, 0.8]$.

$$h(\delta_1, \delta_2) := \begin{cases} 1 - \frac{1}{(1+\delta_1)(1+\delta_2)} & \text{if } \delta_1 + \delta_2 \leq 1 \\ \frac{\delta_1(1-\delta_1^2-\delta_2^2)}{2(1-\delta_1^2)(1-\delta_2^2)} + \frac{\delta_2}{1+\delta_2} & \text{if } \delta_1 + \delta_2 > 1 \end{cases} \tag{C.1}$$

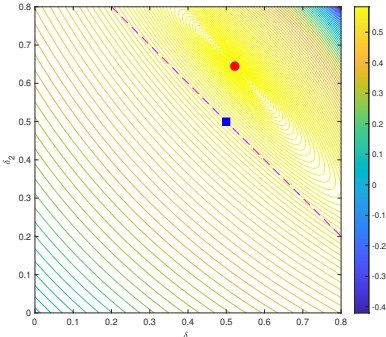

*Figure 6: This contour shows the relation between $\delta_1$, $\delta_2$ and the lower bound of $-\sigma_k L$ presented in (Section 4.1). The dashed line (- - -) shows the border line of the two cases $\delta_1 + \delta_2 = 1$. ($\blacklozenge$) denotes (0.5, 0.5). ($\bullet$) denotes global optimum at $(\hat{\delta}_1, \hat{\delta}_2) \approx (0.52212, 0.644793)$.*

The function is continuous on the dashed line $\delta_1 + \delta_2 = 1$. However, its properties change noticeably across the dashed line. Notably, on the lower left side of the dashed line, the contour lines exhibit symmetry, which vanishes on the upper right side. Moreover, when either $\delta_1$ or $\delta_2$ approaches 1, the function value tends to plummet towards negative infinity.

## C.2 WEAK MVI HALFSPACE AND (MDEG)

In Fig. 1 we give intuitive explanations for projection algorithms involved in this paper. The blue regions represent the weak MVI halfspace generated from the latest iteration point. Fig. 1a explains why GDA fails at star-negative conomonotonic problems, as the hyperplane cannot separate $z^k$ and $z^*$. Fig. 1b shows the principle of (AdaptiveEG+). A larger extrapolation stepsize helps separating $z^k$ and $z^*$, and a smaller update stepsize complete the projection. Fig. 1c demonstrates how (MDEG) works. Instead of increasing stepsize, consecutive extrapolations exploit the structure efficiently.

The example used in the figures is Example 1 with $a = 5$, $b = -1$, and $z^0 = (0, -1)$, $\gamma = -1/2L$, $\sigma_k = 1.2 \cdot \rho$.

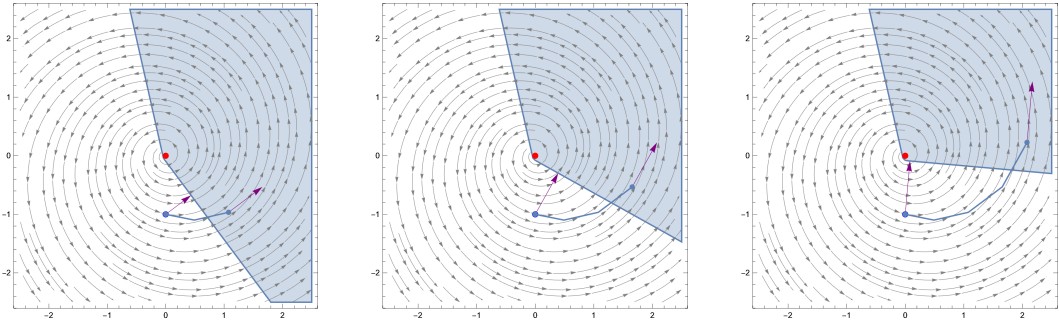

*Figure 7: Projection onto a convex hull*

We mention another perspective that the max distance projection in Algorithm 1 can be considered as an approximation of the projection onto a convex hull. Consider $\delta_k = \rho$ for convenience. Instead of computing $d_i^k = d(z^k, \mathcal{D}(z_i^k))$, construct a convex hull $\bigcap_i \mathcal{D}(z_i^k)$ and compute the distance $d_i^k = d(z^k, \bigcap_{j=1}^{i} \mathcal{D}(z_j^k))$. On obtaining the maximum distance, similarly project onto this convex hull, $\alpha_k = P_{\bigcap_i \mathcal{D}(z_i^k)}(z^k)$. This approach, while computationally more expensive, utilizes all the information available throughout the entire exploration process.

As shown in Fig. 7, hyperplane projection performs a good approximation of the convex hull scheme. This idea could be helpful in exploring other subroutines. For example, it is feasible to choose $z_i^k$ using Monte Carlo method. When employing such desultory and inconsecutive approach, the convex

hull scheme brings considerable improvement over hyperplane projection. See Fig. 8 for an example where the exploration points are randomly selected.

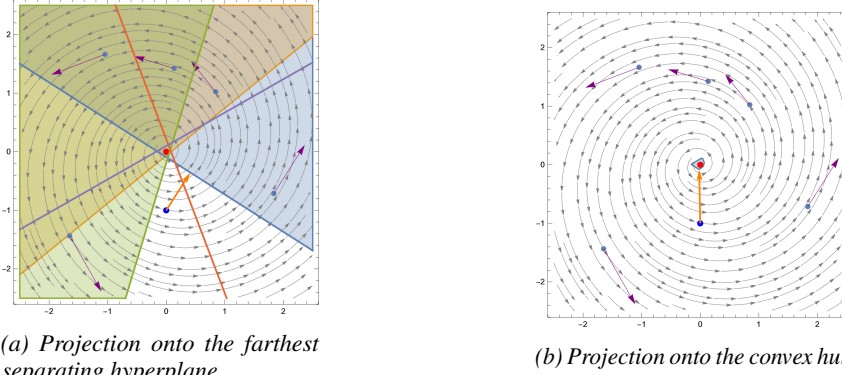

*(a) Projection onto the farthest separating hyperplane*

*(b) Projection onto the convex hull*

*Figure 8: Comparison when randomly choosing exploration points. The orange arrow denotes the adopted projection.*

### C.3 RELATION TO PAST EXTRAGRADIENT

Past Extragradient ((Popov, 1980)) is recently proved to converge for weak MVI problems with $\rho > -1/2L$ ((Choudhury et al., 2023; Gorbunov et al., 2023; Böhm, 2022)).

Past Extragradient has the following form:

$$\bar{z}^k = z^k - \gamma_k F \bar{z}^{k-1}$$
$$z^{k+1} = z^k - \omega_k F \bar{z}^k \tag{C.2}$$

Since the three points $z^{k-1}, z^k, \bar{z}^k$ are collinear, it can be viewed as PEG derives $\bar{z}^k$ directly from $z^{k-1}$ and retraces a step back to get $z^k$. Despite the differences in extrapolation, both AdaptiveEG+ and PEG perform updates in the form of $z^{k+1} = z^k - \omega_k F \bar{z}^k$, suggesting a projection from $z^k$ onto a hyperplane perpendicular to $F \bar{z}^k$.

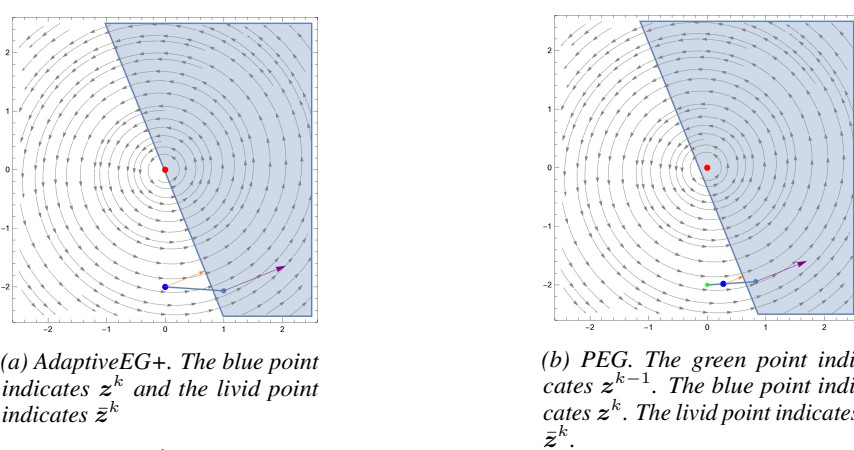

*(a) AdaptiveEG+. The blue point indicates $z^k$ and the livid point indicates $\bar{z}^k$*

.

*(b) PEG. The green point indicates $z^{k-1}$. The blue point indicates $z^k$. The livid point indicates $\bar{z}^k$.*

*Figure 9: Comparison of AdaptiveEG+ and PEG*

Moreover, the underlying principles for convergence are similar:

In AdaptiveEG+, perform a extrapolation step from $z^k$: $\bar{z}^k = z^k - \gamma_k F z^k$, and the weak MVI hyperplane $\partial \mathcal{D}(\bar{z}^k)$ separates $z^k$ and $z^*$.

In PEG, perform an extrapolation step from $z^{k-1}$: $\bar{z}^k = z^{k-1} - (\gamma_k + \omega_{k-1}) F \bar{z}^{k-1}$, and the weak MVI hyperplane $\partial \mathcal{D}(\bar{z}^k)$ separates $z^{k-1}$ and $z^*$.

Note that in AdaptiveEG+ the extrapolation step follows the form of gradient descent, while it is not the case with PEG.

### C.4  $n$-STEP EG AND CARTESIAN OVALS

When we consider n-step extrapolation, we often perceive the last gradient $Fz_n^k$ as benchmark and measure anterior extrapolations with it. This perspective naturally results from the update rule of $\alpha_k$.

When there is 1 extrapolation step, $Fz^k$ is distributed over a circle (ball).

$$\|Fz^k - \frac{1}{1-\delta_1^2}Fz_1^k\| \leq \frac{\delta_1}{1-\delta_1^2}\|Fz_1^k\| \tag{C.3}$$

When there are 2 extrapolation steps, $Fz^k$ is distributed inside a Cartesian oval (surface).

$$\|Fz^k - \frac{1}{1-\delta_1^2}Fz_1^k\| \leq \frac{\delta_1}{1-\delta_1^2}\|Fz_1^k\| \tag{C.4}$$

$$\|Fz_1^k - \frac{1}{1-\delta_2^2}Fz_2^k\| \leq \frac{\delta_2}{1-\delta_2^2}\|Fz_2^k\| \tag{C.5}$$

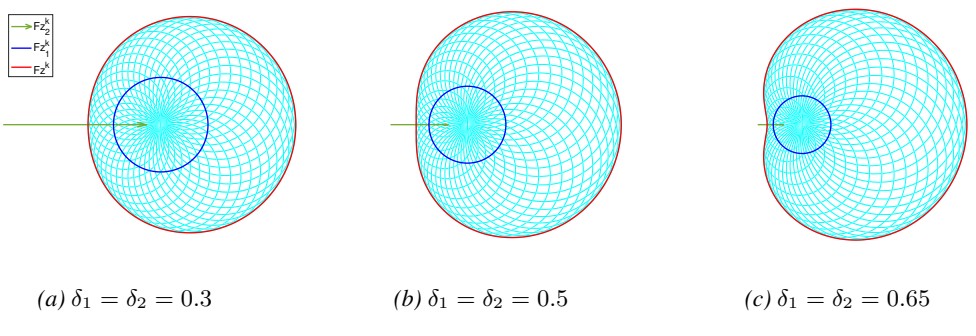

*(a) $\delta_1 = \delta_2 = 0.3$*  *(b) $\delta_1 = \delta_2 = 0.5$*  *(c) $\delta_1 = \delta_2 = 0.65$*

*Figure 10: Cartesian ovals.* $\sqrt{(x - \frac{1}{1-\delta_1^2})^2 + y^2} = \frac{\delta_1}{1-\delta_1^2} + \delta_2\sqrt{x^2 + y^2}$, *where $Fz_2^k = (1, 0)$.*

When there are 3 extrapolation steps, $Fz^k$ is distributed inside the envelope of circles (balls) whose center is inside a Cartesian oval (surface).

$$\|Fz^k - \frac{1}{1-\delta_1^2}Fz_1^k\| \leq \frac{\delta_1}{1-\delta_1^2}\|Fz_1^k\| \tag{C.6}$$

$$\|Fz_1^k - \frac{1}{1-\delta_2^2}Fz_2^k\| \leq \frac{\delta_2}{1-\delta_2^2}\|Fz_2^k\| \tag{C.7}$$

$$\|Fz_2^k - \frac{1}{1-\delta_3^2}Fz_3^k\| \leq \frac{\delta3}{1-\delta_3^2}\|Fz_3^k\| \tag{C.8}$$

We have conducted numerical calculations on 3-step and 4-step cases. The results are presented below along with $n = 1, 2$ cases.

| n | $\delta_1$ | $\delta_2$ | $\delta_3$ | $\delta_4$ | $-\rho_0 L$ |
|---|---|---|---|---|---|
| 1 | 1 | | | | 0.5 |
| 2 | 0.52212 | 0.644793 | | | 0.583456 |
| 3 | 0.272899 | 0.512753 | 0.515522 | | 0.632242 |
| 4 | 0.22 | 0.31 | 0.39 | 0.44 | 0.657724 |

The $n = 3$ result is (nearly) tight, obtained from numerical optimization method in *Mathematica*. The $n = 4$ result is achieved by manual tests and serves as a lower bound. We anticipate that a larger $n$ will continue to yield an improved range for $\rho$.

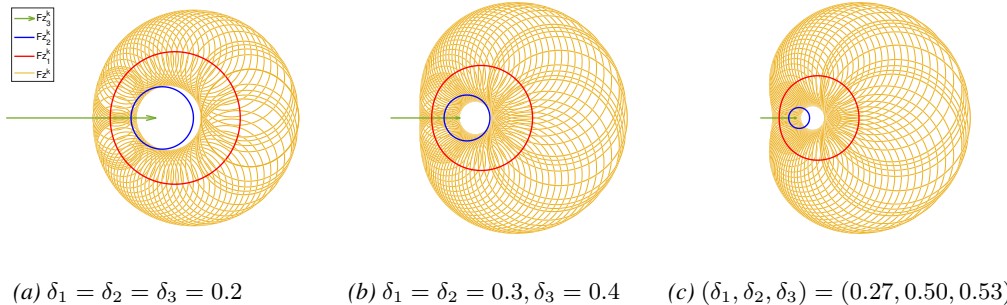

*(a)* $\delta_1 = \delta_2 = \delta_3 = 0.2$   *(b)* $\delta_1 = \delta_2 = 0.3, \delta_3 = 0.4$   *(c)* $(\delta_1, \delta_2, \delta_3) = (0.27, 0.50, 0.53)$

*Figure 11: 3-step EG. Zero curvature is observed again when $\delta_1 + \delta_2 + \delta_3 = 1$.*

# D   ADDITIONAL STATEMENTS

## D.1   EXPERIMENT PARAMETERS

For Example 1, two experiments are conducted.
In the first experiment, $a = 3\sqrt{2}$, $b = -\sqrt{7}$. $\gamma_{k,i} = \frac{1}{2L}, \sigma_k = -\frac{10}{9}\gamma_i \cdot 0.99$ is used in 2-step EG.
In the second experiment, $a = \sqrt{2}$, $b = -1$. $\gamma_{k,i} = \frac{1}{4L}, \sigma_k = -\frac{1476}{625}\gamma_i \cdot 0.99$ is used in 4-step EG.

For Example 2 and Example 3, in both experiment $\gamma_{k,i} = \frac{1}{L}$ and $\sigma_k = -\frac{1}{2L}$ is used in (MDEG).

Recommended tolerance for (MDEG) is $\varepsilon_1 = 10^{-3}$, $\varepsilon_2 = 10^{-3}$.

## D.2   PROPERTIES OF EXAMPLE 1

**Lemma D.1.** *The saddle gradient operator $F$ of $f(x,y) \coloneqq axy + \frac{b}{2}(x^2 - y^2)$, where $a > 0$, $b < 0$, satisfies Assumption 1 with $L = \sqrt{a^2 + b^2}$ and Assumption 2 with $\sigma = \frac{b}{a^2+b^2}$.*

*Proof.* The operator $F\boldsymbol{z} = \begin{bmatrix} \nabla_x f(x,y) \\ -\nabla_y f(x,y) \end{bmatrix} = \begin{bmatrix} ay + bx \\ by - ax \end{bmatrix} = A\boldsymbol{z}$, where the matrix $A \coloneqq \begin{bmatrix} b & a \\ -a & b \end{bmatrix}$.

$$\|F\boldsymbol{u} - F\boldsymbol{v}\| = \|A\boldsymbol{u} - A\boldsymbol{v}\| \geq \|A\|_2 \|\boldsymbol{u} - \boldsymbol{v}\| \tag{D.1}$$

Therefore, the Lipschitz constant $L = \|A\|_2 = \sqrt{a^2 + b^2}$.

Recall the weak Minty condition, where $f(x,y)$ has the only stationary point $\boldsymbol{z}^* = (0,0)$.

$$\langle F\boldsymbol{z}, \boldsymbol{z} - \boldsymbol{z}^* \rangle \geq \rho \|F\boldsymbol{z}\|^2 \tag{D.2}$$

$$(A\boldsymbol{z})^T(\boldsymbol{z} - \boldsymbol{z}^*) \geq \rho(A\boldsymbol{z})^T A\boldsymbol{z} \tag{D.3}$$

$$\boldsymbol{z}^T A^T \boldsymbol{z} \geq \rho \boldsymbol{z}^T A^T A\boldsymbol{z} \tag{D.4}$$

$$b(x^2 + y^2) \geq \rho(a^2 + b^2)(x^2 + y^2) \tag{D.5}$$

Therefore, the weak Minty parameter $\rho = \frac{b}{a^2+b^2}$.   $\square$

We provide an interesting result that (Algorithm 1) converges on Example 1 for $\rho > -\frac{1}{L}$, if the stepsizes are selected infinitely small.

**Theorem D.2.** *Consider Example 1, $f(x,y) \coloneqq axy + \frac{b}{2}(x^2 - y^2)$, where $a > 0$, $b < 0$. Apply (Algorithm 1) on Example 1 with infinitely small stepsize $\gamma_k \to 0$. Assume that $\sigma_k = \rho$, $\lambda_k = 1$, $\boldsymbol{z}^0 = (x_0, y_0)$, then the algorithm converges to the the stationary point after one iteration, i.e. $\boldsymbol{z}^1 = \boldsymbol{z}^* = (0,0)$.*

*Proof.* Recall that $F\boldsymbol{z} = \begin{bmatrix} \nabla_x f(x,y) \\ -\nabla_y f(x,y) \end{bmatrix} = \begin{bmatrix} ay + bx \\ by - ax \end{bmatrix} = A\boldsymbol{z}$, where the matrix $A \coloneqq \begin{bmatrix} b & a \\ -a & b \end{bmatrix}$.
Try to normalize $A$:

$$A = \frac{1}{\sqrt{a^2 + b^2}} \begin{bmatrix} \frac{b}{\sqrt{a^2+b^2}} & \frac{a}{\sqrt{a^2+b^2}} \\ -\frac{a}{\sqrt{a^2+b^2}} & \frac{b}{\sqrt{a^2+b^2}} \end{bmatrix} \tag{D.6}$$

$$= \frac{1}{L} \begin{bmatrix} -\cos\varphi & \sin\varphi \\ -\sin\varphi & -c\cos\varphi \end{bmatrix} \tag{D.7}$$

where $\varphi := \arctan -\frac{a}{b}$. $\cos\varphi = -\frac{b}{\sqrt{a^2+b^2}}$, $\sin\varphi = \frac{a}{\sqrt{a^2+b^2}}$.

When $\gamma_k \to 0$, the discrete process of gradient descent $z_i^k = z_{i-1}^k - \gamma_k F z_{i-1}^k$ evolves into a continuous gradient flow

$$\dot{z} = \frac{dz}{dt} = -Fz \tag{D.8}$$

which corresponds to a linear system $\dot{z} = -Az$. The solution to the ODEs is

$$z(t) = e^{-At} z(0) \tag{D.9}$$

$$= e^{-bt} \begin{bmatrix} \cos at & -\sin at \\ \sin at & \cos at \end{bmatrix} z(0) \tag{D.10}$$

where $z(0) = z^0 = (x_0, y_0) = r_0(\cos\theta_0, \sin\theta_0)$.

Calculate the gradient vector $Fz(t)$,

$$Fz(t) = Ae^{-At} z(0) \tag{D.11}$$

$$= e^{-bt} \begin{bmatrix} b\cos at + a\sin at & a\cos at - b\sin at \\ -a\cos at + b\sin at & b\cos at + a\sin at \end{bmatrix} z(0) \tag{D.12}$$

$$= Le^{-bt} \begin{bmatrix} -\cos(at+\varphi) & \sin(at+\varphi) \\ -\sin(at+\varphi) & -\cos(at+\varphi) \end{bmatrix} z(0) \tag{D.13}$$

Since the matrix is a rotation matrix,

$$\|Fz(t)\| = Le^{-bt}\|z(0)\| = Lr_0 e^{-bt} \tag{D.14}$$

Furthermore,

$$\langle Fz(t), z(t) \rangle = z(t)^T Fz(t) \tag{D.15}$$

$$= -Le^{-2bt} z(0)^T \begin{bmatrix} \cos at & \sin at \\ -\sin at & \cos at \end{bmatrix} \begin{bmatrix} \cos(at+\varphi) & -\sin(at+\varphi) \\ \sin(at+\varphi) & \cos(at+\varphi) \end{bmatrix} z(0) \tag{D.16}$$

$$= -Le^{-2bt} z(0)^T \begin{bmatrix} \cos\varphi & -\sin\varphi \\ \sin\varphi & \cos\varphi \end{bmatrix} z(0) \tag{D.17}$$

$$= -Le^{-2bt}(x_0^2 + y_0^2)\cos\varphi \tag{D.18}$$

$$= -Lr_0^2 e^{-2bt}\cos\varphi \tag{D.19}$$

The matrix multiplication is straightforward since they are rotation matrices.

$$\langle Fz(t), z(0) \rangle = z(0)^T Fz(t) \tag{D.20}$$

$$= -Le^{-bt} z(0)^T \begin{bmatrix} \cos(at+\varphi) & -\sin(at+\varphi) \\ \sin(at+\varphi) & \cos(at+\varphi) \end{bmatrix} z(0) \tag{D.21}$$

$$= -Le^{-bt}(x_0^2 + y_0^2)\cos(at+\varphi) \tag{D.22}$$

$$= -Lr_0^2 e^{-bt}\cos(at+\varphi) \tag{D.23}$$

Note that $\rho L = \frac{b}{\sqrt{a^2+b^2}} = -\cos\varphi$. Now we can calculate the projection distance,

$$d(z(0), \mathcal{D}(z(t))) \tag{D.24}$$

$$=\rho\|F\boldsymbol{z}(t)\| - \frac{\langle F\boldsymbol{z}(t), \boldsymbol{z}(t) - \boldsymbol{z}(0)\rangle}{\|F\boldsymbol{z}(t)\|} \tag{D.25}$$

$$=\rho L r_0 e^{-bt} + r_0 e^{-bt}\cos\varphi - r_0\cos(at+\varphi) \tag{D.26}$$

$$=-r_0 e^{-bt}\cos\varphi + r_0 e^{-bt}\cos\varphi - r_0\cos(at+\varphi) \tag{D.27}$$

$$=-r_0\cos(at+\varphi) \tag{D.28}$$

Let us simplify the notation with $d(t) := d(\boldsymbol{z}(0), \mathcal{D}(\boldsymbol{z}(t)))$. The derivative $d'(t) = ar_0\sin(at+\varphi)$. At the starting point, $d(0) = -r_0\cos\varphi < 0$, $d'(0) = ar_0\sin\varphi > 0$. The signed distance is initially negative and within a increasing interval.

Algorithm 1 adopts the projection when the distance function reaches its first local maximum value. Given the nature of the cosine function, it becomes apparent that the final distance will be $\bar{d}^0 = d(\frac{\pi-\varphi}{a}) = r_0$. Consequently,

$$F\bar{\boldsymbol{z}}^0 = F\boldsymbol{z}\left(\frac{\pi-\varphi}{a}\right) \tag{D.29}$$

$$= e^{-\frac{b(\pi-\varphi)}{a}} L \begin{bmatrix} 1 & 0 \\ 0 & 1 \end{bmatrix} \boldsymbol{z}(0) = e^{-\frac{b(\pi-\varphi)}{a}} L \boldsymbol{z}^0 \tag{D.30}$$

The stepsize $\alpha_0 = \frac{\bar{d}^0}{\|F\bar{\boldsymbol{z}}^0\|} = e^{\frac{b(\pi-\varphi)}{a}} L^{-1}$. Therefore,

$$\boldsymbol{z}^1 = \boldsymbol{z}^0 - \alpha_0 F\bar{\boldsymbol{z}}^0 = \boldsymbol{z}^0 - \boldsymbol{z}^0 = \boldsymbol{0} = \boldsymbol{z}^* \tag{D.31}$$

establishing the theorem.

$\square$

### D.3 PROPERTIES OF EXAMPLE 3

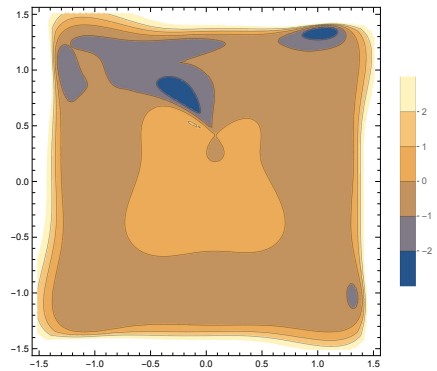

*Figure 12: Contour of local $\rho L$ in Example 3*

It is introduced by Pethick et al. (2022) that in the box $\|\boldsymbol{z}\|_\infty \leq 3/2$, the object function has Lipschitz constant $L = \frac{1}{80}\sqrt{\frac{1}{2}(1089\sqrt{801761} + 993841)} \approx 12.4026$. The critical point is $\boldsymbol{z}^* = (0.0780267, 0.411934)$. According to our calculation, global weak MVI parameter records $-1.52057$ at the point $\boldsymbol{z}' = (-0.258079, 0.791652)$. Recall that $-\frac{1}{L} \approx -0.0806285$.

However, this astonishing value of $\rho L \approx -18.8589$ does not reflect much of its nature. Looking into the local value of $\rho$ and $L$ in Fig. 12 we find that most areas remain tractable parameter of $\rho > -\frac{1}{L}$, while a minimum value of $\rho L \approx -3.04076$ is recorded at the point $(1.03889, 1.35309)$. Furthermore in the outer areas colored in white, $\rho L$ rises dramatically to very large positive value. It is plausible that these anomalies prevent convergence, and convergent algorithms actually circumvent such thorny area and make progress in tractable area.

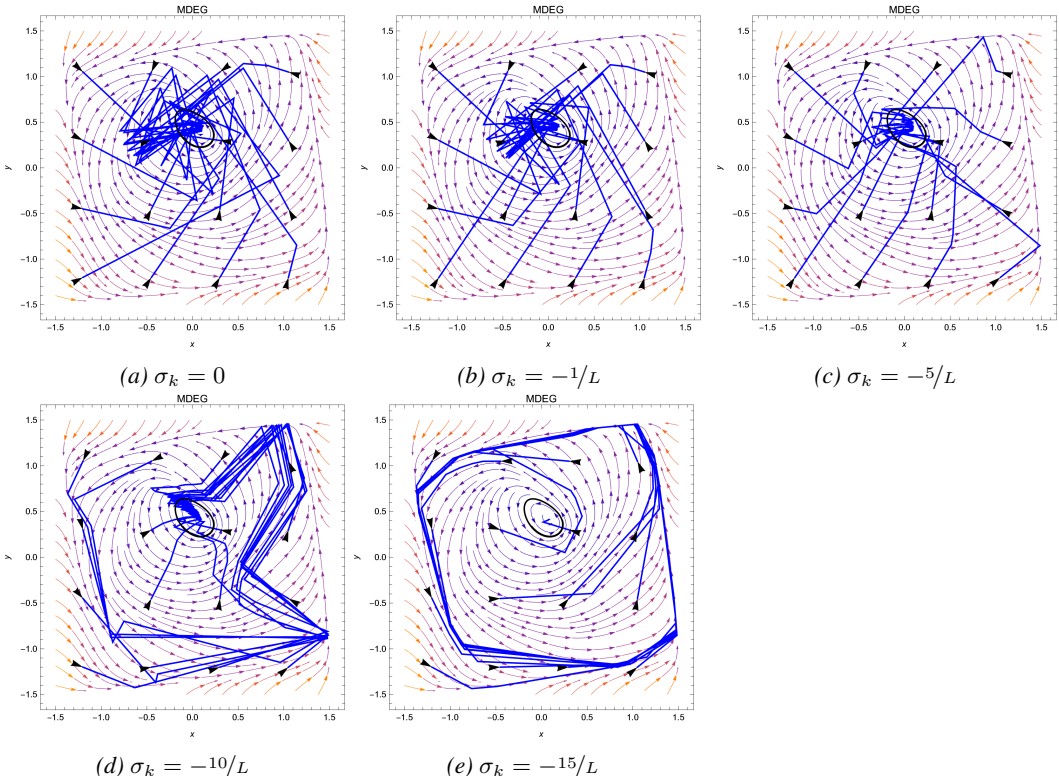

*Figure 13: Parameter adaptability of (MDEG).*

As shown in Fig. 13, a wide range of choice of $\sigma_k$ leads to convergence. From $\sigma_k = 0$ to $\sigma_k = -\frac{10}{L}$, (MDEG) withstands the limit cycles. Convergence to limit cycle is observed under the parameter of $\sigma_k = -\frac{15}{L}$. While the global $L \approx 12.4026$ reflects a spike of local Lipschitz constant near the border, even considering $L_l \approx 1.97703$ in a smaller box of $\|z\|_\infty \leq 1$, the algorithm makes a good coverage of $\sigma_k$ from $-\frac{1}{L_l}$ to $0$.

