# OpenReview forum: "Weaker MVI Condition: Extragradient Methods with Multi-Step Exploration"
_ICLR.cc/2024/Conference — ICLR 2024 poster_

### Official Review · Reviewer_ii6f · 2023-10-31

**Soundness:** 3 good
**Presentation:** 3 good
**Contribution:** 3 good
**Rating:** 8
**Confidence:** 5

**Summary:**

The paper considers the weak Minty variational inequality for unconstrained problems and shows that the range for the problem constant $\rho \in (-1/(2L), \infty)$ can be increase to $\rho \in (-(1-1/e)/L, \infty)$ for an explicit scheme. They achieve this by modifying an extragradient method to compute the extrapolated point using _repeated_ applications of the forward operator. They first generalizing the hyperplane projection approached used in Pethick et al. 2022 to an abstract condition on the extrapolated point. They then show tight conditions on $\rho$ when the extrapolated point is generated by two steps of GDA with time-invariant stepsize. To extending to $\rho < -(1-1/e)/L$ they consider $n>2$ inner steps under the restriction that the sum of stepsizes $\sum_i \gamma_i < 1/L$. They finally propose a method which heuristically attempt to maximize the projection distance in order to relax the requirement on $\rho$.

**Strengths:**

This paper introduces a very nice idea and does so in an approachable way. It is interesting that an explicit scheme can extend the range of $\rho$ and that this is achievable by such a simple and elegant approach. The work opens up exploration of other subroutines.

**Weaknesses:**

I definitely recommend accept. I only have one concern regarding Algorithm 1 and otherwise some suggestions for improving the content/writing.

- It seems problematic that the max distance algorithm (Algorithm 1) has worse complexity when the operator $F$ has _more_ structure (e.g. when cocoercive such that the subroutine itself is sufficient for convergence). In other words, it seems that the scheme is trading off complexity in easier problem for a convergence in a larger class. What is the observed oracle complexity if you run the algorithm on a strongly monotone problem or cocoercive (for which the subroutine convergence) in comparison with e.g. EG+? (e.g. take $f(x,y)=x^2 - y^2 + xy$)

Some suggestions:

- I would state the minimal $n$ that achieves the $\rho_0$ in Thm. 3.4. This way you can also specify the oracle complexity.
- Lemma 2.4 is very related to [Solodov and Svaiter 1999](https://www.emis.de/journals/JCA/vol.6_no.1/j149.pdf) Lemma 2.1 which might be worth mentioning
- Maybe make it explicit that the multiple steps for the extrapolation steps are not "anchored" to approximate a prox ala [Nemirovski 2004, page 3](https://www2.isye.gatech.edu/~nemirovs/SIOPT_042562-1.pdf)
- Concerning $\rho < -1/L$ threshold maybe mention [Bauschke et al. 2019, e.g. Table 1](https://arxiv.org/pdf/1902.09827.pdf) characterization for cohypomonotone problems.
- Page 2 (first paragraph): It is stated that a better bound is derived in the 2-step case, but this seems not to be the case $\rho > -(1-1/e)/L$ is looser than $\rho > -0.5834/L$.
- "common practice is to choose $\sigma_k$ near its lower bound": There is a trade-off (lower bound yields weaker conditions on $\rho$, but higher complexity, so only recommended if guaranteed convergence for the largest class possible is important)
- The discussion after Thm. 3.4 is confusing to me:
    - The lower bound could still be tight for time-invariant stepsize (as Thm. 3.4 is concerned with)
    - The numerical evaluation only _suggests_ that improving the lower bound is possible if time varying stepsize is allowed. Stating that a counterexample proofs something in this setting can be misleading (especially in the context of the word "lower bound" being used in a slightly unconventional way).
- Maybe use "range of $\rho$" instead of lower bound where possible to avoid conflation with negative results (e.g. after Thm. 3.4)

Concerning the writing:

- A lot of whitespace
    - To compress page 6 you could consider writing the inner loop of (n-step EG) with $\forall i \in [n]$ in one line.
- Unspecified in places (e.g. section 2.1 should define $\alpha_k$ and $\gamma_k$, "recommended parameter" requires specification, etc)
- Before section 2 (Preliminaries) the $\rho > -1/L$ should only be a $\rho > -1/(2L)$.
- Assumption 2 is missing $z \in \mathbb R^d$
- page 4 "[...] does not necessarily have to be attained by a _single_ forward operator evaluation"
- $\bar \alpha_k$ is sometimes used for a pre-defined (nonadaptive) stepsize in the literature, but it is up to you what convention you stick to.

**Questions:**

- Eq. 3.6 (and consequently Thm 3.2) is tight for time invariant stepsize? Since this would be a stronger claim I would state it explicitly, and contrast it with e.g. Thm. 3.4.
- Why define $z^0 = z^{init}$?
- Is it possible to prove anything for Algorithm 1?
- After Algorithm 1 it is suggested to use the stepsize choice of n-step EG, but how do you guarantee $\sum_i \gamma_i < 1/L$ in Algorithm 1 if the horizon is not known? (since the horizon is adaptively chosen by the max distance stopping criterion)

---

> ### Author Response · Authors · 2023-11-22
>
> We would like to thank you for your encouraging remarks and instructive review on our work. We address the concerns below.
>
> ##### Weaknesses
>
> - We agree on the point that Algorithm 1 has worse complexity if $F$ is more structured. We believe that it depends on how the subroutine converges to the stationary point. For example on Example 1, where the trajectory of ODEs would be a logarithmic spiral, the subroutine will possibly stop near the next point $v$ on the trajectory that makes $Fv$ parallel to $z^k-z^* $.
>
> - We have conducted several experiments on Example 1, with $a>0, b>0$. The stepsizes for all methods are fixed to $\gamma_k=1/(5L)$. The numerals shown in the sheet denote the operator evaluations the algorithm have taken until $\lVert F\bar{z}^k\rVert<1e-10$. ($\lVert Fz^k\rVert$ for GDA)
>
> - We provide two cases for the selection of $\sigma_k$. In the next sheet, the parameter is set to $\sigma_k = 0$.
>
>   | a    | b    | GDA       | AdaptiveEG+ | 2-step EG | Algorithm 1 |
>   | ---- | ---- | --------- | ----------- | --------- | ----------- |
>   | 50   | 1    | divergent | 2321        | 931       | 242         |
>   | 10   | 1    | divergent | 1289        | 628       | 137         |
>   | 2    | 1    | 328       | 435         | 271       | 162         |
>   | 1    | 1    | 174       | 289         | 199       | 211         |
>   | 1    | 2    | 129       | 241         | 175       | 251         |
>   | 1    | 10   | 118       | 233         | 175       | 338         |
>   | 1    | 50   | 125       | 247         | 184       | 309         |
>
> - The following sheet presents the results when $\sigma_k = \rho$.
>
>   | a    | b    | GDA       | AdaptiveEG+ | 2-step EG | Algorithm 1 |
>   | ---- | ---- | --------- | ----------- | --------- | ----------- |
>   | 50   | 1    | divergent | 2299        | 928       | 252         |
>   | 10   | 1    | divergent | 1137        | 598       | 137         |
>   | 2    | 1    | 328       | 211         | 178       | 93          |
>   | 1    | 1    | 174       | 91          | 91        | 68          |
>   | 1    | 2    | 129       | 47          | 55        | 60          |
>   | 1    | 10   | 118       | 21          | 28        | 32          |
>   | 1    | 50   | 125       | 15          | 19        | 33          |
>
> - It is observed that Algorithm 1 has relatively worse complexity when $\rho$ is close to $1/L$  and has relatively better complexity when $\rho$ is close to 0.
>
> ##### Suggestions
>
> - **On the $n$ that achieves optimal $\rho$** We have included the minimal $n$ in the revised Thm. 4.4 (ii).
> - **On Lemma 2.4** Lemma 2.4 has been relocated to the appendix. We have included remarks in Appendix A to connect the two lemmas.
> - **The $\rho<-1/L$ threshold**  We have included a reference to the specified work in Section 4.
> - **On contribution 1** We have updated our expression to avoid confusion.
> - **The discussion after previous Thm. 3.4 (now Thm. 4.4)**
>   - Even when the stepsizes are time-invariant, there are still better results when $\sum_i \gamma_i>1/L$. In the 2-step case, the tight result is $\rho_0 L\approx-0.582565$ when $\gamma_i\approx0.579785/L$. Refer to the contour for a better understanding (now numbered as Fig. 6).
>   - We have updated the expression to avoid confusion.
>
> ##### Concerning the writing
>
> - **The inner loop of (n-step EG)** We appreciate your suggestion and have implemented the suggested notation.
> - **Unspecified in places** Fixed.
> - **Typos** Fixed.
> - **Usage of $\bar{\alpha}_k$** We have eliminated the symbol and now explicitly present $\lambda_k$ in the update formula.
>
> ##### Questions
>
> - **Tight for time invariant stepsize?** No. There could be improvement after removing the constraint on the sum of stepsizes and we believe that this is true for every $n$.
> - **Why define $z^0=z^{init}$ ?** It is unnecessary. We have removed the definition.
> - **Is it possible to prove anything for Algorithm 1?** We manage to prove that Algorithm 1 with infinitely small stepsize and information of $\rho$ converges on Example 1 with any $a>0$, $b<0$, i.e. $\rho\in(-1/L,0)$. The proof has been included in Appendix D.2. We believe that the result can be generalized. This could be related to the property of Example 1 that the local $\rho$ is same everywhere.
> - **How to guarantee the stepsize constraint in Algorithm 1?** The idea lies in that if the initial $n$ stepsizes are chosen in accordance with Thm 4.4,  it is guaranteed that $d_n^k>0$. Therefore the adopted distance $\bar{d}^k\geq d_n^k>0$. However there has been lacking assurance in (n-step EG) that the distances are always increasing during the subroutine, although we believe it is true. (They do increase from negative to positive, but could there be fluctuations?) Prevent stopping during the initial $n$ steps might constitute potential solution.
>
> We hope these revisions have addressed the concerns raised. We remain open to any further suggestions that might enhance the clarity and comprehension of our paper.

---

> > ### Comment · Reviewer_ii6f · 2023-11-22
> >
> > I thank the authors for thoroughly engaging with the review and will maintain my score. My only comment is to suggest explicitly pointing out the limitation of Algorithm 1. That is, max distance comes at a price, namely in terms of complexity for otherwise "easy" problems.

---

> > > ### Author Response · Authors · 2023-11-23
> > >
> > > We thank you for your swift reply and the favorable reviews. We have integrated your feedback into the latest revision of our paper.

---

> > > > ### Comment · Reviewer_ii6f · 2023-11-23
> > > >
> > > > It came to my awareness that [concurrent work](https://openreview.net/pdf?id=rmLTwKGiSP) achieved the looser requirement $\rho > -1/L$ by also using a hyperplane projection (see e.g. Thm. 4 for an explicit scheme). The "correct" subroutine seems to be the anchored approach used to approximate a proximal update (see e.g.  [Nemirovski 2004, page 3](https://www2.isye.gatech.edu/~nemirovs/SIOPT_042562-1.pdf) as mentioned in the review above). This unfortunately makes it much less interesting to consider _repeated_ applications of the forward operator as done in the paper. I think it is important to at least state that a _different_ subroutine than what is considered in the paper might be desirable.
> > > >
> > > > I will maintain my score, as these are concurrent works, and leave it up to the AC to decide.

---

> ### Author Response · Authors · 2023-11-23
>
> We are grateful for the timely information you provided and have accordingly updated our paper to incorporate the statement. We would still like to contend that their methodologies markedly differ from ours and our contributions remain solid on the improvements and convergence results of the simple techniques of splitting extragradient extrapolation.
>
> Thank you once again for all the informative feedbacks you have provided.

---

> ### Comment · Area_Chair_dBe8 · 2023-12-01
> **Further feedback**
>
> Thank you all for engaging in this discussion.
>
> I first want to relay a comment regarding the concurrent work, as communicated to me by the authors:
>
> > Comment to reviewer ii6f
>
> > We have noticed that in the mentioned concurrent work, the requirement in Thm. 4 is $\rho\in[0,\frac{2}{2L+L_r})$ where $L$ is the Lipschitz constant of the saddle gradient, while their structural assumption is $\langle x-x^*,w\rangle \geq -\frac{\rho}{2}\lVert w\rVert^2$. Therefore their requirement is still stronger than $\rho>-\frac{1}{2L}$ in our notation.
>
> From my own point of view, concurrent works should not influence the decision on whether to accept a paper, but what matters is whether a paper on its own has sufficient contributions to be of value to the broad ML/OPT community. My read of the review ii6f and the discussion is that the reviewer does see value in this work, but please correct me if I am missing something.

---

> > ### Comment · Reviewer_ii6f · 2023-12-03
> >
> > It is in fact the value of the result I question. Let me try to elaborate.
> >
> > The concern is probably more apparent if we look at Theorem 2 of the [concurrent work](https://openreview.net/pdf?id=rmLTwKGiSP) which achieves convergence for $\rho > -1/L$ (in the notation of this Submission2999 under question). Simply pick the euclidean case$h(z)=\frac{1}{2*stepsize}\|z\|^2$ such that $\nabla h(z)=\frac{1}{stepsize} I$.
> >
> > Thm 2 is concerning an (inexact) implicit scheme relying on the resolvent (see D.2 for extension to the inexact case), but you can easily recover an explicit scheme by using the gradient method for the strongly-monotone and Lipschitz subproblem (see e.g. https://openreview.net/pdf?id=b1JPBGJhUi). In the notation of Submission2999 the resulting scheme would correspond to using an "anchored" update
> >
> > $G_k(z) = Q_z \circ Q_z \circ \cdots \circ Q_z(z)$  where $Q_z(w)=z - \gamma F(w)$
> >
> > in Equation 3.1 of Submission2999 instead of the forward operator
> >
> > $G_k = (\operatorname{id} - \gamma F)\circ(\operatorname{id} - \gamma F)\circ \cdots \circ(\operatorname{id} - \gamma F)$
> >
> > So it appears that by approximating the resolvent/prox you can achieve $\rho > -1/L$, but the method suffers a log factor (due to the subsolver). It suggests that one should attempt to approximate the prox explicitly and that $\operatorname{id} - \gamma F$ for the inner step is simply a poor approximation.
> >
> > It is still interesting though, that it is possible to shave off the log factor. In the light of the above, the primary contribution seem to be that _the log factor can be shaved off_ even beyond $\rho < -\frac{1}{2L}$ (otherwise EG+ would suffice) as long as $\rho > -(1-1/e)/L$. The exact form of inner steps $n$ becomes important (the authors did include the expression in their rebuttal, but I would maybe suggest highlighting the oracle complexity under contributions).

---

> > > ### Comment · Area_Chair_dBe8 · 2023-12-04
> > > **Another comment from the authors**
> > >
> > > I am relaying another message I have received from the authors:
> > >
> > > We greatly appreciate your attention and explanations.
> > >
> > > Upon reexamining the concurrent work, we offer our interpretation below, adapted to our notation.
> > >
> > > For the euclidean case $h=\frac{L_h}{2}\lVert\cdot\rVert^2$, the BPP iteration reduces to the proximal point algorithm $z^{k+1}=z^k-\frac{1}{L_h}Fz^{k+1}$. Theorem 2 of that work demonstrates that this proximal update works for all $F$ satisfying $\rho>-1/L_h$.
> > >
> > > The problem lies in the relationship between $L$ and $L_h$. We find it hard to assume $L_h$ close enough to $L$, since $L=L_h$ implies that $Fz^k=0$.
> > > $$
> > > \lVert Fz^{k+1}-Fz^k \rVert \leq L\lVert z^{k+1}-z^k \rVert =\frac{L}{L_h}\lVert Fz^{k+1} \rVert
> > > $$
> > > Interestingly, this inequality also appears in our analysis. For EG+ it holds when subtituting $L/L_h$ with 1/2. And for n-step EG, it holds with 1-1/(1+1/n)^n).
> > >
> > > Thus, we believe that the critical issue remains in the gap between $L$ and $L_h$. It seems to depend on the specific approach of approximating the prox method. The realization in the concurrent work (SA-GDmax with projection) attains the existing $\rho>-1/(2L)$ since there are constraints between $\tau$, $\rho$ and $L$. Additionally, the constraints $\tau<1/(L+\hat{L})$ and $\tau>\rho/(2-\rho L)$ looks very similar to our expression $-\rho/(1+\rho L)<\gamma_k\leq 1/L$ in Section 3.
> > >
> > > We appreciate the idea of approximating prox method and will cover this perspective in further revision. Your engagement in this discussion has significantly deepened our understanding. We welcome any corrections if you find errors in our reply.

---

> > > > ### Comment · Reviewer_ii6f · 2023-12-04
> > > >
> > > > Let me try to make my argument clearer by first writing the construction in the notation of the current work (and then only subsequently translating to the notation of the authors).
> > > >
> > > > Thm. 2 of the concurrent work only requires $\rho L_h < 2 \Leftrightarrow L_h < 2/\rho$. This is achieved by
> > > >
> > > > $\nabla h(x)=\frac{2}{\tau} x$ with $\tau > \rho$.
> > > >
> > > > for which the Lipschitz modulus is $L_h = 2/\tau$ since trivially
> > > >
> > > > $\|\nabla h(x)-\nabla h(y)\| = \frac{2}{\tau} \|x-y\|.$
> > > >
> > > > What is a valid choice of $\tau$? We need to (efficiently) approximate the resolvent:
> > > >
> > > > $(I + \frac{\tau}{2}M)^{-1}$
> > > >
> > > > It suffice for $\frac{\tau}{2}M$ to be a contraction. From Lipschitz of $M$ we have:
> > > >
> > > > $\|\frac{\tau}{2}M(x)- \frac{\tau}{2}M(y)\| \leq \frac{\tau L}{2} \|x-y\|$
> > > >
> > > > So we just need $\tau/2 < 1/L$.
> > > >
> > > > Now in the notation of the authors $F=M$ and the stepsize requirement reduces to:
> > > >
> > > > $\gamma =\tau/2 <1/L$
> > > >
> > > > and the condition on $\rho$ (in the notation of the authors) which I will rename to $\hat p$ to avoid confusion:
> > > >
> > > > $1/\gamma =2/\tau < 2/\rho = -1/\hat \rho$
> > > >
> > > > In other words sufficient condition reduces to:
> > > >
> > > > $\hat \rho > -\gamma > -1/L$

---

### Official Review · Reviewer_xeFY · 2023-10-31

**Soundness:** 3 good
**Presentation:** 3 good
**Contribution:** 3 good
**Rating:** 6
**Confidence:** 3

**Summary:**

This paper extends the AdaptiveEG+ proposed in [1] and studies the extragradient-type algorithms that take multiple intermediate steps for the unconstrained weak MVI problem, aiming to relax the weak MVI condition. Their analysis essentially relies on the condition that the iteration point and the solution point can be separated by the border hyperplane defined by the weak MVI condition w.r.t. the iteration point. In detail, for the iterate $z_k$, the authors defines the adaptive stepsize for computing $z_{k + 1}$ to be $\lambda_k(\rho \|F(z_k)\|^2 - \langle F(z_k), z_k - \bar z_k\rangle)/\|F(\bar z_k)\|^2$ with $\lambda_k \in (0, 2)$ and certain intermediate iterate $\bar z_{k}$ (not necessarily obtained by one GD step), which requires $\bar z_k$ to satisfy the condition that $\rho \|F(z_k)\|^2 > \langle F(z_k), z_k - \bar z_k\rangle$, assuming that the operator $F$ satisfies weak MVI condition with (negative) parameter $\rho$. To guarantee this, the authors derives certain conditions on the stepsizes of intermediate step for multi-step extragradient, which leads to wider range on weak MVI condition parameter, roughly $\rho > -\frac{1 - 1/e}{L}$ (not tight), and improves upon the previously best known range $\rho > -1/2L$. Their findings are further supported by preliminary numerical results.


[1] Thomas Pethick, Panagiotis Patrinos, Olivier Fercoq, Volkan Cevhera, et al. Escaping limit cycles: Global convergence for constrained nonconvex-nonconcave minimax problems. In International Conference on Learning Representations, 2022.

**Strengths:**

1. The authors provides an intuitive generalization of AdaptiveEG+ from the perspective of weak MVI halfspaces. As long as one can find some intermediate iterate $\bar z_k$ such that $\rho \|F(z_k)\|^2 > \langle F(z_k), z_k - \bar z_k\rangle$, the algorithm framework (2.9) are proved to converge with rate $O(1/k)$ for the average guarantee. An adaptive exploration scheme to find such $\bar z_k$ is also given in Algorithm 1.
2. This paper gives a quite comprehensive analysis and discussion on the stepsize requirements for intermediate steps with connection to the restriction on the parameter of the weak MVI condition, relaxing the previous contraints $\rho > -1/2L$ to roughly $\rho > -(1 - 1/e)/ L$.

**Weaknesses:**

I do appreciate the discussion and analysis for the more general multi-step extragradient. However, I note that the convergence rate remains the same (see Thm 2.5 in the paper) as AdaptiveEG+[1] in prior works, and the wider range of $\rho$ only provides slight (constant) improvements. One may argue that although the rate is the same, MDEG and multi-step EG can solve for problems in more nonmonotonic regions. However, the numerical experiment as in Figure 4 shows that CurvatureEG+[1] can actually perform similarly as MDEG proposed in this paper. All of these limit the contribution of the paper.

**Questions:**

1. Could the authors further explain the derivation for the inequality in (3.1)?
2. Besides the adaptive exploration proposed in Algorithm 1, I am curious for $n$-step EG whether it is expected to get better range on $\rho$ with larger $n$, or it is sufficient to only take $n = 3$ with proper intermediate stepsizes for better range on $\rho$? Based on the current (non-tight for $n \geq 3$) analysis in the paper, one can get better range on $\rho$ with 3 steps than taking $n$ steps with $n$ tends to infinity (in which case we get $\rho > -(1 - 1/e)/L$).

---

> ### Author Response · Authors · 2023-11-22
>
> We thank the reviewer for reviewing our paper and for the insightful comments. We address the specific concerns below.
>
> **Weaknesses**
>
> - We are grateful for the author's positive acknowledgment of our analysis. We would like to clarify that CurvatureEG+ uses local curvature information from linesearch, while the proposed methods only involve global information about the operator. This fundamental difference underpins the potential of our methods. We suggest that integrating local information could further enhance the efficacy of our approach.
>
>   Additionally, the concept of a generalized framework, as introduced in our paper, not only serves the immediate purpose of addressing the problem at hand but also lays the groundwork for future research. We believe that this framework opens up new avenues for exploration.
>
> **Questions**
>
> 1. Eq. 3.1, now renumbered as Eq. 4.1, originates from AdaptiveEG+. We have introduced this algorithm in Section 2.1. It may looks different from the presentation in the original work ([1], Alg. 1, 1.2). We would like to mention that it is helpful to replace the notation $H=id-γ_kF$ using $H\bar{z}^k-Hz^k=-\gamma_k F\bar{z}^k$ (which is true in unconstrained situations).
>
> 2. We anticipate that a larger $n$ will yield an improved range for $\rho$. The constraint of $\sum_i \gamma_i L<1$ represents a compromise, given the complexity of the otherwise condition. Furthermore, stepsize choice $(\gamma_1,\gamma_2)$ in the 2-step scenario can be regarded as special cases $(\gamma_1,\gamma_2,0)$, $(\gamma_1,0,\gamma_2)$, $(0,\gamma_1,\gamma_2)$ in the 3-step scenario. Consequently, we expect that expanding the number of dimensions will enhance the overall results.
>
>    We have conducted new numerical calculations to determine the stepsizes that minimize $\rho_0 L$ under the scenarios of $n=3$ and $n=4$ . We enumerate the results below, covering the $n=2$ case as well.
>
>    |      | $\delta_1$ | $\delta_2$ | $\delta_3$ | $\delta_4$ | $-\rho_0 L$ |
>    | ---- | ---------- | ---------- | ---------- | ---------- | ----------- |
>    | n=2  | 0.52212    | 0.644793   |            |            | 0.583456    |
>    | n=3  | 0.272899   | 0.512753   | 0.515522   |            | 0.632242    |
>    | n=4  | 0.22       | 0.31       | 0.39       | 0.44       | 0.657724    |
>
>    The result for $n=3$ is derived by the numerical maximize function *NMaximize[]* in Mathematica and is tight. The result for $n=4$ is achieved by manual tests and serves as a lower bound. These results confirm our expectations.
>
>    The codes used for the calculations have been added to the supplementary materials. (`3stepEG.nb` and `4stepEG.nb`)
>
> [1] Thomas Pethick, Panagiotis Patrinos, Olivier Fercoq, Volkan Cevhera ̊, et al. Escaping limit cycles: Global convergence for constrained nonconvex-nonconcave minimax problems. In International Conference on Learning Representations, 2022.

---

> > ### Comment · Reviewer_xeFY · 2023-11-23
> > **Request for further clarifications on Eq. 4.1**
> >
> > I thank the authors' reply. I still have questions on the derivation on Eq. 4.1. As authors suggest, we have $H\bar{z}^k - Hz^k = -\gamma_k F\bar{z}^k$, leading to
> > $$
> > \langle Fz^k, F\bar{z}^k \rangle = \langle -\frac{1}{\gamma_k}(\bar z^k - z^k), -\frac{1}{\gamma_k}(H\bar{z}^k - Hz^k) \rangle = \frac{1}{\gamma_k^2}\langle \bar z^k - z^k, H\bar{z}^k - Hz^k \rangle.
> > $$
> > Could the author further explain why $H$ is $1/(1 + \gamma_k L)$-cocoercive? Also, based on the inequality the authors had as Eq. 4.1, we should have $\langle Fz^k, F\bar{z}^k \rangle \geq 0$, no? Then I am wondering how the authors apply Cauchy-Schwarz inequality? It seems the inequality direction is incorrect.
> >
> > I appreciate the efforts for revising the paper. However, I suggest that the authors can move the newly added part in Section 3 to the introduction with a new subsection for discussing the motivation/context and certain techniques.
> >
> > I would like to maintain my score for now, **but I wish to receive the authors' response on Eq. 4.1**. I would like to keep my score in the margin due the limited contribution on the complexity aspect, and the restricted analysis beyond the $n = 2$ case (for both $n$-step EG and Alg. 1).

---

> ### Author Response · Authors · 2023-11-23
>
> Thank you for your prompt feedback. We address the questions below.
>
> 1. **On different representations of the algorithm**
>
>    In (AdaptiveEG+),
>    $$
>    \alpha_k=\frac{\delta_k}{\gamma_k}-\frac{\langle H\bar{z}^k-Hz^k, \bar{z}^k-z^k\rangle}{\lVert H\bar{z}^k-Hz^k\rVert^2}=\frac{\delta_k}{\gamma_k}-\frac{\langle F\bar{z}^k, \bar{z}^k-z^k\rangle}{\gamma_k\lVert F\bar{z}^k\rVert^2} $$
>    $$
>    z^{k+1}=z^k+\lambda_k\alpha_k(H\bar{z}^k-Hz^k)=z^k-\lambda_k\alpha_k\gamma_kF\bar{z}^k
>    $$
>    which is equivalent to our representation. Note that there is difference in the definition of $\alpha_k$.
>
> 2. **On the inequality in Eq. (4.1)**
>
>    We provide the detailed proof below.
>    $$
>    \lVert Fz^k-F\bar{z}^k \rVert\leq \gamma_kL\lVert Fz^k \rVert $$
>
>    $$
>    (1-\gamma_k^2L^2)\lVert Fz^k\rVert^2+\lVert F\bar{z}^k\rVert^2-2\langle Fz^k, F\bar{z}^k\rangle \leq 0$$
>
>    $$
>    \lVert Fz^k\rVert^2-\frac{2}{1-\gamma_k^2L^2}\langle Fz^k, F\bar{z}^k\rangle +(\frac{1}{1-\gamma_k^2L^2})^2 \lVert F\bar{z}^k\rVert^2 \leq \frac{\gamma_k^2L^2}{(1-\gamma_k^2L^2)^2}\lVert F\bar{z}^k\rVert^2$$
>
>    $$
>    \lVert Fz^k-\frac{1}{1-\gamma_k^2L^2} F\bar{z}^k\rVert \leq \frac{\gamma_kL}{1-\gamma_k^2L^2}\lVert F\bar{z}^k\rVert$$
>
>    $$
>    \langle F\bar{z}^k, Fz^k-\frac{1}{1-\gamma_k^2L^2} F\bar{z}^k\rangle \geq-\frac{\gamma_kL}{1-\gamma_k^2L^2}\lVert F\bar{z}^k\rVert^2 $$
>    $$
>    \langle F\bar{z}^k, Fz^k\rangle \geq \frac{1}{1+\gamma_kL}\lVert F\bar{z}^k\rVert^2
>    $$
>    A similar process can be found in the proof of Lemma A.2, Eq. (A.17)~(A.22).
>
>    We acknowledge that it would be somewhat hasty to summarize this as "follows from $\lVert F\bar{z}^k-Fz^k\rVert \leq \gamma_k L \lVert Fz^k\rVert$ and Cauchy-Schwarz inequality". We will refine the expression in the subsequent revision. (done)
>
> 3. We value your advice regarding Sec. 3 and will consider restructuring in future revisions.
>
> We hope these responses address your question and thank you once again for your valuable attention and feedback.

---

### Official Review · Reviewer_mxmH · 2023-11-03

**Soundness:** 3 good
**Presentation:** 1 poor
**Contribution:** 2 fair
**Rating:** 5
**Confidence:** 4

**Summary:**

The paper focuses on the weak MVI condition and proposes algorithms that allow the condition to be weaker than what was analyzed before. In particular, the main idea is that by using algorithms with multiple exploration steps, one can further relax prior used assumptions. In addition, the authors design an adaptive algorithm that explores until the optimal improvement is achieved. This process exploits information from the whole trajectory and effectively tackles cyclic behaviors.

**Strengths:**

The paper is well-written, and the idea is easy to follow. The authors did a great job in the introduction of motivating the problem, explaining their approach, and clearly stating their main contributions.

**Weaknesses:**

However, I believe the paper has several issues regarding the presentation of the sections after the Introduction that make the reader's life more challenging.  The narrative around the statements of the theorems and the figures in the experiments section is not clear. In general i find that the main part of the paper has several holes and requires an effort from the reader to connect the dots.

Let me provide some details below:

### Some Basic Issues:

1)  Figure 1 is included in the paper without reference to it at any part of the paper. What is its purpose? Can the authors elaborate more? The figure mentions "range of gradient: while the definitions are related to operators. Does this hold only for the specific operator at the end of page 2 (saddle gradient operator)? Why is this figure helpful for the reader?  In several parts of the paper, the authors refer to the "gradient". If the results hold for general operators, it would be beneficial to be mentioned.

2)  What is the $S^c$ in equation 2.6?

3) Proof of Lemma 2.3 is missing (even if it is trivial the authors should include it for completeness).

4) The presentation of Lemma 2.4, its importance, and the discussion after the results should be improved as this is a vital part of the understanding of the paper. I would suggest the authors include more information and explain the motivation behind it.  For example, what is the purpose of parameter $\lambda_k$ here (why is not simply equal to 1)? Also, how does algorithm 2.9 satisfy the assumptions of Lemma 2.4? More explanation is needed.

5) in section 3 the authors mentioned, "The key is to calculate stepsize from projection and assure its positiveness." I get the statement but then how is that related to what they have as equation (3.1)? important connections are missing. Why is that projection?

### More Serious Presentation Issues in the main part of the paper :

6)  Sections 3.1 and 3.2 are focused on the algorithms and statements of the theorems. However, there is no explanation of what the results mean and how they are positioned in the literature. What are the benefits of the final statements? As a reader, I find it impossible to understand why for example, Theorem 3.2 is useful and what it means. The whole section 3 and in particular, pages 5 and 6 include several theorems one behind the other without a proper presentation and paragraphs explaining in more detail the main outcome.

7) Section 4 is devoted to the Max Distance Extragradient where the extrapolation process will not stop until the projection distance stops increasing. This is interesting, but an intuitive understanding of what this means is missing from the main paper. The Discussion from Appendix B.1 and Figure 6 should have been part of the main paper.

8) Important: An indication of the lack of proper presentation is the following. There is no clear explanation in the main paper of why by taking multiple extrapolation steps, one can have a weaker MVI. This is the main selling point of the work, but all the discussion of this is in the appendix. By simply reading the main paper, this is not clear.

9) Unfortunately, the same pattern happening in the experiments section as well. The problems are presented, but an in-depth comparison of the results and paragraphs focusing on the output of the proposed methods and how they are compared to other algorithms is missing. How do the experiments correspond to the main theorems? A full page is devoted only to the plots (Figures 3-5) but without proper presentation and connection to the theoretical results.

### On related work

9) Finally, I believe the cited work of Choudhury et al. 2023 includes convergence of the deterministic optimistic methods for solving weak Minty inequalities with $\rho>-1/2L$. It might be worth comparing with the Optimistic method as well in experiments and in the comparison of theory in the main paper.

**Questions:**

See Weaknesses section above.

---

> ### Author Response · Authors · 2023-11-22
>
> We would like to thank the reviewer for your conscientious and exhaustive review. The constructive feedback we received has greatly facilitated the enhancement of our paper.
>
> We address the concerns as follows:
>
> ##### Basic Issues
>
> 1. We have moved Fig. 1 to Appendix C.1 due to space limitation and added explanatory text along with corresponding algorithms. The plots are simple geometric representations of Assumption 1 & 2, so they hold for any vector.
>
>    We have updated the figure to include our new result of $\rho>-(1-1/e)/L$ and have refrained from using "gradient" in the figure legend. This figure is helpful in that it presents a visual understanding for weak Minty conditions. For example, it presents a straightforward explanation on why MVI ($\rho=0$) is a powerful condition and why $\rho\leq-1/L$ situations are intractable.
>
>    Concerning the usage of the term "gradient" , we agree that it is more accurate to avoid it here. However, in subsequent sections, its usage seems inevitable, as we employ various operators in a manner analogous to gradients, in line with the term "extragradient".
>
> 2. $\mathcal{S}^c$ denotes the complement of the set $\mathcal{S}$. We have included this notation in the definition.
>
> 3. We appreciate the reviewer's reminder to include the proof of Lemma 2.3, which has now been incorporated into the revised manuscript.
>
> 4. We have moved the previous Alg. 2.9 and Thm. 2.5 to an independent section and elaborated on the motivation. We have relocated Lemma 2.4 to Appendix A, as it serves as an intermediary result in our analysis.
>
>    $\lambda_k$ is a conventional scaling parameter in related algorithms. The idea lies in that scaling the projection distance with a factor in $(0,2)$ still progresses toward the target. This parameter provides flexibility, encompassing any small enough update stepsize into the algorithm. In the initial manuscript, we consistently set $\lambda_k$=1. In the revised paper we have incorporated $\lambda_k$ into algorithm presentation and theorems for completeness.
>
> 5. We have elaborated the idea in the newly added Section 3. The concept starts from (AdaptiveEG+) where the update are interpreted as a projection onto a hyperplane. We have as well introduced the link between calculation of $\alpha_k$ and the projection distance.
>
> ##### More Serious Presentation Issues
>
> 6. We have restructured the sections (now numbered 4.1 and 4.2), along with the theorems. Each theorem now is accompanied by explanatory remarks that elucidate their significance and utility. Guiding text has been added to both sections to ensure clarity and prevent any potential confusion among readers.
> 7. We appreciate your advice and have moved the figure to Section 3, along with a detailed explanation.
> 8. It is challenging to explain the underlying idea. We have introduced the perspective in Section 3 that exploration provides new information and narrows down the possible region of $z^*$. Another related viewpoint has been supplemented in Appendix C.2.  We hope that these efforts will enhance the clarity and comprehension of our work.
> 9. We have added a paragraph on Page 8 to elucidate the result of the experiments.
>
> ##### On related work
>
> 10. We thank you for your suggestion regarding optimistic methods. We find this idea very interesting. Due to space limitation, we have included a concise analysis in Appendix C.3 that interpret the algorithm from the perspective of projection.
>
>     Regarding the experiments, we believe that single-call algorithms represent a different direction in the performance-complexity tradeoff compared to our algorithm, which is why we chose not to include them in the comparison.
>
> We hope these revisions have addressed the concerns raised. We remain open to any further suggestions that might enhance the clarity and comprehension of our paper.

---

### Official Review · Reviewer_xJXM · 2023-11-07

**Soundness:** 3 good
**Presentation:** 2 fair
**Contribution:** 2 fair
**Rating:** 5
**Confidence:** 3

**Summary:**

The paper focuses on solving structured, non-monotone variational inequality problems that satisfy the weak Minty inequality (eq. 2.3). The proposed approach is based on multi-step methods (per iteration), where multiple evaluations of the operator are computed at consecutive points extrapolated through an iterative subroutine at each parameter update.
It is demonstrated that an extragradient method with two or finite extrapolation steps can solve this problem, provided that the problem's parameters are known. Additionally, a variant is proposed that adaptively determines the number of extrapolation steps required, relying on knowledge of a parameter of the assumed problem structure.

**Strengths:**

The paper addresses a challenging problem, establishes convergence guarantees in a unifying way, and provides some intuition. The first method slightly modifies existing ones, whereas the second one (Alg. 1) builds on the theoretical results.

**Weaknesses:**

*The guarantee requires knowing $\rho$*.
To ensure convergence, it is necessary to know the $\rho$ problem parameter. However, this presents a challenge as the convergence guarantee depends on it. Although the authors suggest ways to estimate it, it remains uncertain if the proposed approach will converge when using a noisy estimate. This limitation also applies to Alg. 1, which also requires knowledge of $\rho$ to function correctly.



*The presentation of the theorems*.
The theorems in the second part are presented in a very convoluted manner, making it difficult to interpret the added parameters. Some of the included assumptions are not evident from the statements, which further adds to the confusion. After carefully re-reading the second part, it is still unclear what the precise assumptions, convergence rate, and convergence measure are.


*Writing.*
The writeup needs further refinement. See the non-exhaustive examples below (listed as Minor).


*Additional practical concerns.*
The studied methods involve a variable number of inner steps, which makes it more complex than the methods discussed in this paper. As a result, I am uncertain whether it can be applied to a wider range of problem classes. However, it is reassuring that it offers guarantees for the specific problem class addressed in this paper.



# Other

- Many of the figures are not referenced, making it unclear when the reader should consult them. Moreover, these are not explained, for example, it would be helpful to describe how Fig. 2 helps understand the theorems.
- I wouldn't classify it as a *more than local* method (page 1) because the information is still very much local, but it's applied to multiple points, which has obvious computational disadvantages.
- $\rho$ should be listed as input to Alg. 1.


# Minor



- The abstract has numerous typos, for example:
   - Min-max problem -> The min-max problem
   -  application in ML ->  applications in ML
   - exhaustive researches -> exhaustive research
   - problem on nonconvex-nonconcave setting -> problem of nonconvex-nonconcave setting
   -  is proved -> has proved
   - process exploit  [...] tackle -> process exploits [...] tackles

- Introduction:
   - $\rho, L$ are mentioned without introducing them. It would be helpful if at least (2.3) is moved in the introduction and if it is mentioned that $L$ is the Lipschitz constant
   - typo: algorithm that exploit -> exploits
   - similarly, it was unclear what is $e$ at the beginning of page 2
   - In the first paragraph of page 2, since the comment *this lower bound (on $\rho$) is not necessarily tight* is for two cases (1) when the sum of the step sizes at one update step is smaller than $1/L$ and (2) when that is not the case, it remains very unclear if it is tight for the case (1), because that comment states *that does not meet the step size restriction*. In other words, this statement is problematic because two different cases are being compared while claiming that the bound for one case is not satisfied.
    - The second paragraph on page 2 also assumes that the reader is familiar with the mentioned idea; otherwise, it is unclear why *this algorithm tackles cyclic behaviors*. Moreover, that is also the case for standard extragradient, so it is unclear what the point is here.
- In Fig. 1, it would be helpful to annotate $L$ constant.
- page 3: instead of the *objective function* $F$, it is better to use *vector field* or *operator*
- page 3: our methods [...] is -> are
- page 3: described next section, missing *in*
- ...

**Questions:**

1. On page 4, you mentioned that the lower bound can be represented by parameters in $G_k$. However, I am unsure how convergence can be proven if $\rho$ is estimated as commented therein. Could you please elaborate on this?
2. In Thm. 2.5 what is the assumption on $\rho_0$?
3. Could you describe in more detail how Fig. 2 was obtained? (are $L,\rho$ from a specific problem?)

---

> ### Author Response · Authors · 2023-11-22
>
> We thank the reviewer for your detailed feedback on our paper and address all concerns below.
>
> ##### Weaknesses
>
> - **The guarantee requires knowing $\rho$** We would like to clarify that our methods maintains the same level of requirement for the knowledge of $\rho$ as previous methods. In the revised version, the estimation parameters are incorporated into the theorem and convergence has been proved. Furthermore, the presence of $\lambda_k \in (0,2)$ in the framework accommodates any sufficiently small update stepsize, reducing the dependence of the knowledge of $\rho$ for calculating $\alpha_k$. Alg. 1 has also been revised to include the estimation parameter and Thm. 4.4(i) works for Alg. 1 as well.
>
> - **The presentation of the theorems** We have extensively revised the presentation of the theorems to make them self-contained. The restructured theorems stand independently and do not refer to each other. Moreover, we have incorporated the parameters $\sigma_k$, $\lambda_k$ into the algorithm representation and the theorems. The convergence rate and convergence measure have been clarified in the revised text.
> - **Writing** We have fixed the mentioned issues and refined the remaining part of the paper.
>
> - **Additional practical concerns** Concerning the variable exploration steps, we would like to mention that fixing $n=2$ provides guarantee for $\rho>-0.55/L$ and $n=6$ provides guarantee for $\rho>-0.6/L$. These are significant improvements, achieved with increasing computational cost by a constant factor.
>
> ##### Other
>
> - **Figures not referenced** We have now ensured that each figure in the document is not only referenced but also accompanied by a detailed explanation. Regarding the previous Fig. 2, we've transferred it to Appendix C.1 and provided discussion there. Furthermore, we have reformulated the theorem statements in Section 4.1, enhancing the figure's understandability.
> - **On the expression "more than local"** We agree that for n-step EG the information is still very much local. However, we believe it is not applicable to Alg. 1. In the case of circular limit cycles or divergent GDA, the subroutine often explores through about 1/4 of the circumference. Fig. 1(c) and Fig. 8 (random exploration) may also provide support for this claim. Therefore we would like to stand by this expression.
>
> ##### Minor
>
> - **Typos in abstract and introduction** We thank the reviewer for carefully pointing out these errors. We have fixed them in the revised text.
> - **Missing introduction for parameters** The mentioned parameters are introduced in the revision.
> - **On contribution 1** We have updated our expression to avoid confusion.
> - **On why the algorithms are better**: While it is challenging to clarify the underlying reasons in the contributions section, we have incorporated insights into the new Section 3 to foster a better understanding. We hope this will help mitigate any confusion.
>
> ##### Questions
>
> - **Representing range of $\rho$ by parameters in $G_k$** We have elaborated this issue in the revised paper. The parameters have now been included in Thm. 4.4 in a rigorous manner. The main idea of the proof lies in that parameters within this range fulfill the conditions in Thm 4.3, which in turn are pivotal for ensuring convergence. Additionally, we manage to recover the parameter range of (AdaptiveEG+) when n=1.
> - **Assumption on $\rho$ in the general framework** There is no explicit assumption on $\rho$. The addressable range of $\rho$ depends on the structure of the subroutine. We have covered this issue in the revised paper (Page 3, below Thm. 3.1) and guide readers towards Section 4 for details.
> - **Derivation of Fig. 2** No specific example is involved. This figure is used to visually display the results of Theorems 4.1 and 4.2. The figure is a contour of a bivariate function, now presented in equation (C.1). The Matlab code used to create this figure have been included in the supplementary materials (`2stepEG_range.m`).
>
> We hope these revisions have addressed the concerns raised. We remain open to any further suggestions that might enhance the clarity and comprehension of our paper.

---

### Meta-Review · Area_Chair_dBe8 · 2023-12-05

**Metareview:**

The paper studies a class of non-monotone variational inequalities (VIs) that satisfy a "weak MVI" condition, which was introduced in the recent literature as a condition under which non-monotone VIs with Lipschitz operators can be approximated in a computationally efficient manner (general nonmonotone VIs with Lipschitz operators are computationally intractable). Following the recent line of work, the paper considers extragradient (EG)-type methods. The main insight in the paper is that the acceptable range of the weak MVI parameter $\rho$ can be extended from $\rho > -1/(2L)$ to $\rho > -(1 - 1/e)/L$ by using a repeated forward operator in the extrapolation step. This is nice, because the result shows that we can handle a somewhat larger class of nonmonotone VIs. On the other hand, the result would have been more impressive if it managed to extend the range to $\rho > -1/L$, which we expect to be the limit of what is achievable, based on the results regarding the resolvent of comonotone operators. I would be more positive if there were no other issues related to the presentation of the results and knowledge of the parameter $\rho$, as pointed out by the first two reviews, but I overall think that the paper makes a solid contribution to this line of work.

The paper suffered from various presentation issues that were, at least to some extent, handled in the rebuttal phase. While it is commendable that the authors improved the presentation significantly in the interim, it is generally not a good practice to submit papers that are not quite yet ready and I would recommend that in the future the authors make a more concerted effort to submit polished versions of their work.

**Justification For Why Not Higher Score:**

As I wrote above, this is a borderline paper. I am ok with it going in either direction.

In the revised form, the paper is interesting enough to accept, but I would also be ok with rejecting it, given the issues pointed out by the first two reviews. Only one of the reviewers is very positive, but still has reservations that came up in the discussion phase, so what was written by the reviewer is actually not well reflected by the numerical score.

**Justification For Why Not Lower Score:**

See the comments above.

---

### Decision · Program_Chairs · 2024-01-16

Accept (poster)